# VRPRM: Process Reward Modeling via Visual Reasoning

## Abstract

Process Reward Model (PRM) is widely used in the post-training of Large Language Model (LLM) because it can perform fine-grained evaluation of the reasoning steps of generated content. However, most PRMs lack long-term reasoning and deep thinking capabilities. On the other hand, although a few works have tried to introduce Chain-of-Thought (CoT) capability into PRMs, the annotation cost of CoT-PRM data is too expensive to play a stable role in various tasks. To address the above challenges, we propose VRPRM, a process reward model via visual reasoning, and design an efficient two-stage training strategy. Experimental results show that using only 3.6K CoT-PRM Supervised Fine-Tuning(SFT) data and 50K non-CoT PRM Reinforcement Learning (RL) training data, VRPRM can surpass the non-thinking PRM with a total data volume of 400K and achieved a relative performance improvement of up to 118% over the base model in the BoN experiment. This result confirms that the proposed combined training strategy can achieve higher quality reasoning capabilities at a lower data annotation cost, thus providing a new paradigm for PRM training with more efficient data utilization.

## 1 Introduction

Reward Models (RMs) are a core component in the post-training process of Large Language Models (LLMs) through Reinforcement Learning with Human Feedback (RLHF). However, most current reward models are Outcome Reward Models (ORMs) that are oriented towards evaluating the final result. They can only provide a holistic score for the entire generated content, making it difficult to supervise the critical reasoning steps and internal logical structure of the generation process. As a result, they fail to provide stable reward signals about the quality of the reasoning chain during reinforcement learning.

Therefore, an increasing number of Process Reward Models (PRMs) have been proposed to directly score each step of the generated content. Yet, they face a critical problem: how can a reward model that lacks reasoning ability itself be used to guide a thinking policy model?

To address the poor capability and generalization of reward models, many works on Chain-of-Thought Reward Models (CoT-RMs) have been proposed. As shown in Fig 1, the vast majority of these are CoT-ORM models, with only a few study Zhao et al. (2025) training a PRM by synthesizing CoT-PRM supervised fine-tune (SFT) data, which rely on manual annotation or costly distillation methods. This data bottleneck has become a key obstacle hindering the improvement of PRM performance and generalization across multiple tasks and scenarios.

RL presents a promising approach to not only address the data cost problem but also enhance generalization capabilities beyond what supervised fine-tuning can achieve Chen et al. (2025a); Chu et al. (2025). As shown in the Fig 2, previous studies typically used outcome-level data for reinforcement learning, where result-based rewards encourage the model to guess the correct answer, enabling easy reward through guessing. In contrast, training with process-level data requires evaluating the entire process, with higher scores awarded only for correctly predicting all steps. This reduces the reliance on random guesses, promoting more accurate and structured process evaluation.

In this paper, we propose Visual Reasoning PRM (VRPRM), a first visual PRM with CoT capability, and we design an efficient two-stage training data leveraging strategy. First, supervised fine-tuning (SFT) is performed using a small amount of high-quality CoT-PRM data to activate the model's

| Reward Model | PRM | MM | CoT | RL |
|---|:---:|:---:|:---:|:---:|
| RRM Guo et al. (2025) | | | ✓ | |
| RM-R1 Chen et al. (2025a) | | | ✓ | ✓ |
| Think-RM Hong et al. (2025) | | | ✓ | ✓ |
| R1-Reward Zhang et al. (2025a) | | ✓ | ✓ | ✓ |
| UnifiedReward Wang et al. (2025c) | | ✓ | ✓ | ✓ |
| Qwen-Math-PRM Zhang et al. (2025b) | ✓ | | | |
| GenPRM Zhao et al. (2025) | ✓ | | ✓ | |
| VisualPRM Wang et al. (2025b) | ✓ | ✓ | | |
| **VRPRM (ours)** | ✓ | ✓ | ✓ | ✓ |

Figure 1: The comparison of different RMs. Our VRPRM is the first multi-model PRM with advanced reasoning capabilities enhanced through RL scaling. **MM** represents whether the RM is multi-modal. **CoT** represents whether the RM has thinking capability. **RL** represents whether reinforcement learning is used when training the model.

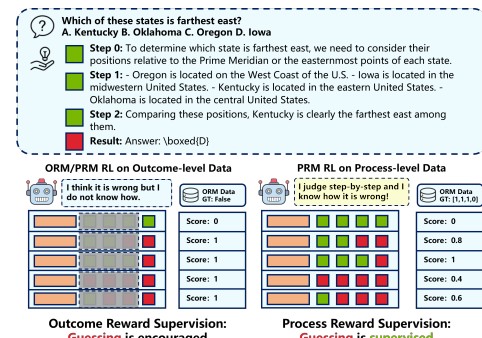

Figure 2: Process-level supervision provides a discriminative RL reward signal. However, under ORM-reward supervision, even when guessing at random, the model still maintains a 50% probability of being rewarded.

initial long-term reasoning and process evaluation capabilities; then, non-CoT PRM data is used to perform reward verification in reinforcement learning, reducing the demand for CoT-PRM data and further enhancing the model's deep thinking ability. Experimental results show that using only 3.6K CoT-PRM SFT data and 50K non-CoT PRM RL training data, VRPRM can surpass the non-thinking PRM with a total data volume of 400K. This result confirms that the proposed combined training strategy can achieve higher quality reasoning capabilities at a lower data annotation cost, thus providing a new paradigm for PRM training with more efficient data utilization.

Our contributions can be summarized as follows:

- **Pioneering the Integration of CoT RL in Visual PRMs.** We are among the first to systematically address the need for deep thinking in PRMs. We introduce VRPRM, the first-ever multimodal CoT-PRM trained by RL, explicitly designed to enhance the fine-grained reasoning and evaluation capabilities of reward models.

- **A Data-Efficient Two-Stage Training Strategy.** This method demonstrates remarkable data efficiency, enabling our model to surpass a traditional PRM trained on 400K data while using less than one-eighth of that amount (specifically, 3.6K CoT-PRM and 50K non-CoT PRM data).

- **A Novel and Effective Test-Time Scaling Approach.** Our VRPRM also serve as a highly effective test-time scaling strategy. It achieves significant performance improvements across multiple multimodal benchmarks, yielding a relative gain of up to 118% over the base model and substantially outperforming current state-of-the-art (SOTA) methods. This showcasing a new avenue for scaling model capabilities.

## 2 RELATED WORK

**Process Reward Models.** Process reward models (PRMs) are playing an increasingly critical role in reinforcement learning (RL) optimization and test time scaling (TTS). Unlike traditional Outcome Reward Models (ORMs) Whitehouse et al. (2025); Wang et al. (2025d;a); Zhang et al. (2024a), which assign a single score to the final output, PRMs evaluate each intermediate step in the generation process. These step-level signals are then aggregated into a final reward, providing more detailed supervision and reducing the issue of "spurious correctness," where a model reaches the correct answer through flawed reasoning. This enables PRMs to show better generalization and stability in complex reasoning tasks. Qwen-Math-PRM Zhang et al. (2025b) combines Monte Carlo estimation with large language model judgments to filter and select a large set of process-level annotated data for supervised fine-tuning. VisualPRM Wang et al. (2025b) uses the InternVL2.5 model series to generate solution steps, applying Monte Carlo sampling to assess step-level accuracy. The model is trained by discretizing the output space into specific tokens. In summary, these studies mainly rely on process-level annotated data for fine-tuning foundation models, giving them process

evaluation capabilities. However, these PRMs lack deep reasoning abilities and struggle to capture the logical structures underlying complex reasoning paths.

**Chain-of-Thought Reward Models.** In recent years, research in reward modeling has shifted from traditional scalar scoring models to Chain-of-Thought Reward Models (CoT-RMs), which generate reasoning chains to assist in preference judgment. RRM Guo et al. (2025) treats reward modeling as a reasoning task, using long-chain reasoning before generating the final reward, and introduces a reinforcement learning (RL) framework to enhance reasoning ability. Many CoT-ORM studies follow a two-stage training approach: first, supervised fine-tuning (SFT) for initialization, and then RL to further improve performance. RM-R1 Chen et al. (2025a) and Think-RM Hong et al. (2025) use high-quality long-chain reasoning data to guide the model via SFT and apply RL to improve performance in the second stage. Later work extended CoT-ORM to multimodal settings. R1-Reward Zhang et al. (2025a) uses GPT-4o to annotate a multimodal dataset and applies RL to enhance performance on complex reward tasks. UnifiedReward-Think Wang et al. (2025c) combines multimodal preference data with RL to improve reasoning across text and images. The CoT approach is also used in Process Reward Models (PRMs), like GenPRM Zhao et al. (2025), which uses explicit CoT reasoning and code verification but does not apply RL. CoT-enhanced reward models improve interpretability and generalization, but they require high-quality CoT data, which is costly to acquire and annotate.

## 3 METHODOLOGY

### 3.1 PROBLEM FORMULATION

In this section, we introduce the preliminary setting of our research problem. Let $\mathcal{D} = \{(I, P, S)\}$ denote a dataset consisting of a problem $P$, image $I$, and solution $S$. Each solution is composed of multiple steps, denoted as $S = (s_1, s_2, \ldots, s_n)$, where $s_i$ represents the $i$-th step.

**Visual PRM.** In VisualPRM Wang et al. (2025b), in order to effectively utilize the generation capability of MLLM, the process evaluation is regarded as a multi-round dialogue, and the probability value predicted by token 1 is used as the score of the step. Let $M$ is a visual prm. Formally, the output of the PRM can be represented as:

$$y_i \sim M(1|I, P, s_{\leq i}), \tag{1}$$

where $y_i$ denotes the score of $i$-th step. By setting a threshold to determine whether the step is correct.

**Visual Reasoning PRM.** By equipping Visual PRM with an explicit reasoning process such as CoT Wei et al. (2022), we have Visual Reasoning PRM. Before evaluating a step, we assume that the model's thinking about a problem $P$, image $I$, and solution $S$ is $\mathcal{T}$, then the output of VRPRM is,

$$\mathcal{R} \sim \pi_\theta(I, P, (s_1, s_2, \ldots, s_n), \mathcal{T}), \tag{2}$$

where $\mathcal{T} \sim \pi_\theta(I, P, (s_1, s_2, \ldots, s_n))$, we extract the formatted output $\mathcal{R}$ to obtain process reward $(r_1, \ldots, r_n)$.

### 3.2 COLD START ACTIVATION CAPABILITY

Although instruction-tuned LLMs have strong generalization capabilities and can complete basic process evaluation tasks through prompts, these models often find it difficult to stably generate structured and parsable evaluation results without cold start. Specifically, the model may not be able to return evaluation results in the expected format, the process evaluation cannot be aligned with the actual number of steps. Therefore, in this section, our main purpose is to stimulate the model CoT and process evaluation capabilities. It mainly includes two parts: (1) synthesis of high-quality CoT-PRM data and (2) SFT based on CoT-PRM data.

#### 3.2.1 SYNTHETIC CoT-PRM DATA

VisualPRM400K Wang et al. (2025b) is a dataset of multimodal reasoning data with process label. We select data that is easy for the model to think and reason about, including science, geometry, functions, physics, biology and other fields. We select about 10K data, each of which

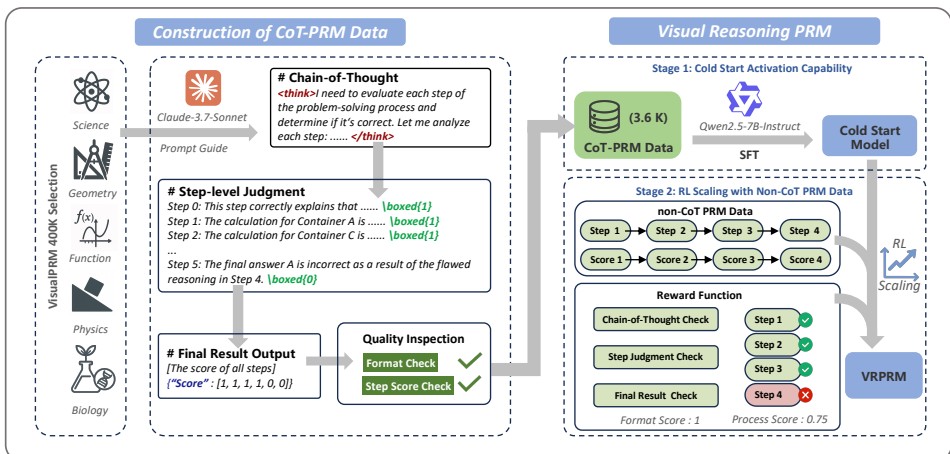

Figure 3: Overall framework of VRPRM. We first use Claude-3.7-Sonnet to generate CoT-PRM data with long-horizon reasoning on a small amount of VisualPRM400K data. **Two-stage training pipeline:** (1) **Cold Start:** We use CoT-PRM data to fine-tune the base model, helping it learn basic thinking and process evaluation capabilities. (2) **RL Scaling:** Then we use non-CoT PRM data to perform RL fine-tuning, further strengthening the model's process evaluation and reasoning capabilities.

contains a prompt $P$, a step-by-step solution $S = (s_1, \ldots, s_n)$, and a process-level annotation $G_r = (g_1, \ldots, g_n)$. Therefore, we can use a LLM to construct evaluation data with long-horizon reasoning and process-level annotations. In this study, we choose Claude-3.7-Sonnet as the data generator.

As shown in Fig 3, to ensure that the data is clearly structured and labeled consistently, we design a systematic prompting strategy that includes the following key steps: Step 1, we guide the model to conduct thinking part to fully understand the problem background, image information and the requirements of the evaluation task. The model's thinking content needs to be placed between <think>and </think>tokens. Step 2, we then guide the model to perform a fine-grained analysis of each solution step and annotate the correctness of each step in a unified format, in the form of \boxed{1} (correct) or \boxed{0} (incorrect). Step 3, the model must also return the intermediate results of the evaluation process in a standardized JSON format, such as {"Score":[$r_1, \ldots, r_n$]};

Based on the above process, we build a batch of PRM data with clear structure and complete long-horizon reasoning. For each generated sample, we implement a strict data quality inspection process to ensure the format specification and label consistency; all data that did not strictly follow the specified format output or the evaluation results deviated from the reference label were eliminated. We finally obtained a dataset containing about 3.6K high-quality question-answer pairs, with a positive-negative sample ratio of about 1:1. For detailed prompt and statistics, please see the Appendix A.

### 3.2.2 SUPERVISED FINE TUNING

We use the above high-quality data to perform SFT on the target model to help the model master basic long-horizon reasoning and initial process assessment capabilities. Its training objectives are defined as follows:

$$r_\theta = \arg \min_{\boldsymbol{\theta}} \mathbb{E}_{(I,P,S,C)\sim\mathcal{D}_{SFT}}[-\log P_{\boldsymbol{\theta}}(C|I, P, S)], \tag{3}$$

Where $\mathcal{D}_{SFT}$ is a constructed CoT-PRM dataset, $P$ is the problem, $S$ is the candidate solution, and $C$ is the target output, including the chain-of-thinking, step-level judgement, and final result output.

### 3.3 RL SCALING WITH NON-COT PRM DATA

To further enhance the model's evaluation ability, we recommend reinforcement learning of the fine-tuned model $r_\theta$ on step-level annotated data. We directly use the fine-tuned process reward model $r_\theta$ as the policy model for optimization, and its objective function is as follows:

$$\max_{r_\theta} \mathbb{E}_{(I,P,S,G_r)\sim\mathcal{D}_{prm},O\sim r_\theta(I,P,S)} [\mathcal{R}(G_r, O)] - \beta\mathbb{D}_{\mathrm{KL}}(r_\theta\|r_{\mathrm{ref}}) \tag{4}$$

Where $r_{ref}$ is the reference reward model. In practice, we use the checkpoint before RL training as $r_{ref}$, that is, the model checkpoint obtained after fine-tuning. $I, P, S$ represents the image, problem, and solution extracted from the data $\mathcal{D}_{prm}$, $G_r = (g_1, \ldots, g_n)$ represents the step-level annotation result, and $O$ represents the text generated by the reward model, which includes the thought chain and process judgment and result output. $\mathcal{R}(G_r, O)$ is the reward function, and $\mathbb{D}_{KL}$ is the KL divergence. In practice, we use Group Relative Policy Optimization (GRPO) Shao et al. (2024) to optimize the objective in the formula.

### 3.3.1 REWARD FUNCTION DESIGN

The rule-based reward mechanism has proven effective in enhancing the model's reasoning ability. In our approach, we design two reward rules when using step-level annotated data for RL: format compliance and process accuracy.

First, the model output must follow a predefined format, which we regard as a reflection of the model's basic evaluation capabilities. Specifically, the model output should contain the following structural elements: the <think>... </think>token for the thought chain, the \boxed{0 or 1} used for step-by-step judgment, and the JSON format output of the final evaluation result, including {"Score":[...]}. The existence of these tokens facilitates the structured extraction of the model's evaluation results. Therefore, if the model does not follow the format requirements, its format reward will be set to zero:

$$\mathcal{R}_{format}(O) = \text{has\_think}(O) \wedge \text{has\_step\_judge}(O) \wedge \text{has\_final\_judge}(O) \tag{5}$$

Since this reward primarily prevents format violations, we assign it a lower weight, as our main focus during the RL stage is improving the model's evaluation capability rather than format adherence.

While format compliance reflects basic output skills, we further introduce process accuracy to evaluate each step of the model's reasoning. This reward is based on the accuracy of the model's predictions for each step. If the final judgment is incorrect, the process reward is set to zero:

$$\mathcal{R}_{process}(G_r, O) = \begin{cases} 0, & \text{if } 1[g_o = r_o] = 0; \\ \dfrac{1}{n} \displaystyle\sum_{i=1}^{n} 1[g_i = r_i], & \text{otherwise.} \end{cases} \tag{6}$$

Here, $1[\cdot]$ is the indicator function, $g_o$ is defined by the process annotation $G_r$ (as in Eq 7), and $r_o$ is defined by the process reward extracted from $O$, similar to $g_o$.

$$g_o = \begin{cases} 0, & \text{if } 0 \in G_r; \\ 1, & \text{otherwise.} \end{cases} \tag{7}$$

The final reward function is,

$$\mathcal{R}(G_r, O) = w_f * \mathcal{R}_{format} + w_p * \mathcal{R}_{process} \tag{8}$$

Where $w_f$ and $w_p$ correspond to the weights of $\mathcal{R}_{format}$ and $\mathcal{R}_{process}$ respectively. In the work we set $w_f = 0.1$ and $w_p = 0.9$.

### 3.4 TEST-TIME SCALING

We follow VisualPRM's setup for BoN Wang et al. (2025b), we set the critic model as a Process Reward Model (PRM) to select the best response from multiple candidate responses.

In the inference phase, PRM scores the generation process of each response step by step: for a response $S = (s_1, s_2, \ldots, s_n)$, we let the PRM model predict the next token at each position and use the probability of token "1" as the reward for that step. Formally, the reward score at each step is defined as:

$$r_t = P_\theta(1|x, s_{<t}) \tag{9}$$

where $x$ is the input prompt, $s_{<t}$ represents the previous $t - 1$ steps. For the $N$ candidate responses $\{S_1, S_2, \ldots, S_N\}$ generated by the model, we input each candidate response into PRM for process scoring and obtain the corresponding average score. Finally, the response with the highest score is selected as the output through the following formula:

$$S = arg \max_{S_i \in \{S_1, S_2, \ldots, S_N\}} \frac{1}{n} \sum_{t=1}^{n} P_\theta(1|x, s_{<t}^i). \tag{10}$$

Table 1: **VisualProcessBench results reported with FEI and AEI. Bold** indicates the best result, underlined indicates the second best result. w/o CoT means VRPRM does not perform explicit reasoning, w/o RL means VRPRM does not perform RL training.

| Model Name | # Samples | MMMU | | MathVision | | MathVerse-VO | | DynaMath | | WeMath | | FEI Avg. | AEI Avg. |
|---|---|---|---|---|---|---|---|---|---|---|---|---|---|
| | | FEI | AEI | FEI | AEI | FEI | AEI | FEI | AEI | FEI | AEI | | |
| **Proprietary Models** | | | | | | | | | | | | | |
| GPT-4o-mini | unk | 40.45 | 35.27 | 27.39 | 35.10 | 28.36 | 34.44 | 40.35 | 37.46 | 45.70 | 37.30 | 36.45 | 35.91 |
| Gemini-2.0-Flash | unk | 43.07 | 43.04 | 30.48 | 40.68 | 36.16 | 40.89 | 55.79 | 43.25 | 52.92 | 42.99 | 43.68 | 42.17 |
| **Open-source Models** | | | | | | | | | | | | | |
| InternVL2.5-8B | unk | 41.63 | 49.59 | 30.67 | 42.61 | 42.96 | 43.62 | 48.88 | 51.24 | 55.33 | 43.35 | 43.89 | 46.08 |
| Qwen2.5-VL-7B | unk | 44.57 | 46.88 | 36.94 | 39.54 | 46.69 | 42.75 | 52.81 | 52.89 | 60.82 | 44.76 | 48.37 | 45.36 |
| Qwen2.5-VL-72B | unk | 46.44 | 51.31 | 34.27 | 41.88 | 42.50 | 45.92 | 51.75 | 53.25 | 57.73 | 46.74 | 46.54 | 47.82 |
| MiMo-VL-7B | unk | 50.94 | 58.45 | 38.27 | 61.60 | 48.71 | 66.15 | 60.50 | 66.81 | 57.05 | 63.87 | 51.09 | 63.38 |
| VisualPRM-8B | 400K | 30.71 | 59.01 | 24.58 | 62.91 | 24.56 | 60.93 | 30.00 | 62.08 | 18.21 | 60.22 | 25.61 | 61.03 |
| **Ours** | | | | | | | | | | | | | |
| VRPRM | 53.6K | 52.06 | 63.16 | 42.98 | 67.34 | 40.94 | 63.80 | 53.51 | 67.95 | 59.11 | 67.76 | 49.72 | 66.00 |
| - w/o CoT | 53.6K | 46.44 | 52.66 | 26.83 | 51.95 | 34.80 | 54.72 | 41.05 | 53.06 | 41.58 | 55.90 | 38.14 | 53.66 |
| - w/o RL | 3.6K | 47.57 | 55.94 | 33.99 | 61.82 | 43.96 | 62.43 | 52.46 | 63.08 | 50.86 | 67.30 | 45.77 | 62.11 |
| - w/o RL & w/o CoT | 3.6K | 49.06 | 50.69 | 33.15 | 54.57 | 41.72 | 51.70 | 50.18 | 55.26 | 48.80 | 48.79 | 44.58 | 52.20 |
| VRPRM-MiMo | 53.6K | 53.18 | 66.95 | 46.07 | 72.87 | 47.95 | 71.93 | 59.47 | 74.55 | 66.32 | 78.26 | 54.60 | 72.91 |
| - w/o CoT | 53.6K | 54.31 | 65.52 | 43.40 | 69.99 | 45.42 | 71.05 | 59.12 | 73.20 | 62.89 | 78.34 | 53.03 | 71.62 |
| - w/o RL | 3.6K | 50.26 | 57.63 | 39.36 | 60.51 | 48.46 | 61.87 | 60.04 | 61.67 | 55.73 | 61.38 | 50.77 | 60.61 |
| - w/o RL & w/o CoT | 3.6K | 55.06 | 45.35 | 35.81 | 44.24 | 46.69 | 46.55 | 59.65 | 50.48 | 60.82 | 47.22 | 51.61 | 46.77 |
| VRPRM-Qwen3 | 53.6K | 52.81 | 65.78 | 42.13 | 70.48 | 44.93 | 70.19 | 57.02 | 72.76 | 62.20 | 70.85 | 51.82 | 70.01 |
| - w/o CoT | 53.6K | 46.44 | 62.34 | 39.89 | 70.15 | 38.99 | 67.23 | 53.33 | 71.85 | 57.04 | 71.57 | 47.14 | 68.63 |

# 4 EXPERIMENTS

In this section, we aim to answer the following questions:

- **Q1:** How does the performance of VRPRM compare to previous PRMs?

- **Q2:** How does VRPRM benefit policy model test-time scaling?

- **Q3:** Can VRPRM effectively exploit CoT reasoning to improve its performance?

## 4.1 EXPERIMENT SETTINGS

**Base Model.** We followed the setup of VisualPRM Wang et al. (2025b), selecting Qwen2.5-7B-Instruct, MiMo-VL-7B-SFT-2508 and Qwen3-4B-VL-Thinking as the initial base model. MiMo-VL-7B-SFT-2508 and Qwen3-4B-VL-Thinking are multimodal models with reasoning capabilities. We first performed SFT to give the model preliminary process scoring capabilities and obtained Cold Start Model. Then we performed RL training on it to strengthen the model capabilities and generate VRPRM, VRPRM-MiMo and VRPRM-Qwen3.

**Benchmarks.** We chose VisualProcessBench Wang et al. (2025b), a widely used multimodel process reward model evaluation benchmark. Each test example in the dataset contains a problem, a step-by-step solution, and a step-level label that reflects whether each step is correct or not. Following the setup of VisualPRM Wang et al. (2025b), we evaluate the best-of-N results of our VRPRM on five benchmarks: MathVista Lu et al. (2024), MathVision Wang et al. (2024), MathVerse Zhang et al. (2024b), WeMath Qiao et al. (2024), and LogicVista Xiao et al. (2024), which will be described in Appendix B.

**Training Settings.** In the SFT stage, for all base models the LoRA rank was set to 16 with an alpha value of 32, the learning rate was $1.0e^{-4}$, and the model was fine-tuned for 3 epochs. We set the per-device batch size to 1 and used 4 gradient accumulation steps. In the RL stage, we use verl Sheng et al. (2024) as our training framework. We train for 2 episodes using the AdamW optimizer with a learning rate of $1.0e^{-6}$ and KL penalty with a coefficient of $1.0e^{-6}$. The RL training operated with a global batch size of 512. For Qwen2.5-VL-7B and MiMo-VL-7B-SFT-2508, we use four 80GB NVIDIA A800 GPUs for SFT and eight for RL. For Qwen3-4B-VL-Thinking, we use NVIDIA H200 GPUs with the same settings.

**Evaluation Metrics.** Inspired by Wang et al. (2025b); Zheng et al. (2024), we use the First Error Identification (FEI) and All Error Identification (AEI) to evaluate the performance of the PRM process evaluation. FEI requires the PRM to identify the first error encountered during reasoning. AEI assesses the PRM's ability to identify all errors in a given solution. Both of them are calculated by F1 scores. This comprehensive error identification is crucial for providing fine-grained rewards during training, enabling effective reinforcement learning. We also record the computation overhead on MMMU benchmark as in table 2, including tokens per sample and processing time. The full computation overhead statistics on VisualProcessBench is in Appendix C.

## 4.2 VISUALPROCESSBENCH RESULTS

**Performance Analysis.** Table 1 shows the performance of the PRM model on VisualProcessBench, where **VRPRM significantly outperforms all existing methods, including both proprietary and open-source models**. Specifically, VRPRM-7B-MiMo and VRPRM-7B-Qwen lead across all sub-datasets. VRPRM-7B-MiMo achieves an average AEI of 66.44 and an average FEI of 51.54, outperforming the leading multimodal PRM, VisualPRM, by 5.41 in AEI and 25.93 in FEI. VRPRM-7B-Qwen also shows improvements of 4.97 in AEI and 24.11 in FEI, despite using only 13.4% of the training data. This highlights the effectiveness of our combined training scheme in boosting performance while keeping data requirements low.Notably, VRPRM without RL (VRPRM-7B-Qwen w/o RL and VRPRM-7B-MiMo w/o RL), trained on just 3.6K samples, achieved strong average AEIs of 62.11 and 60.61, outperforming all other open-source and proprietary models except MiMo-VL-7B. However, VRPRM-7B-MiMo w/o RL showed a slight performance drop compared to its base model, indicating that the initial SFT phase may have partially disrupted its CoT structure. Nevertheless, the subsequent RL training phase helped to recover this gap. See the Ablation Analysis for more details, and Appendix C for responses to VRPRM.

**Computational Overhead Analysis.** As shown in Table 2, the primary computational overhead stems directly from generating detailed reasoning process. For instance, VRPRM generates an average of 339.73 output tokens per sample compared to 10.03 tokens per sample for the non-reasoning baseline, resulting in an inference time increase from 0.39s to 16.94s per sample.

Table 2: **Computation overhead analysis on VisualProcessBench(MMMU).** Metrics include average token counts and processing time per sample/reward.

| Model | Total Tokens / Sample | Output Tokens / Sample | Input Tokens / Sample | Time / Sample (s) | Time / Reward (s) |
|---|---|---|---|---|---|
| VRPRM | 2305.82 | 339.73 | 1966.08 | 16.94 | 1.5353 |
| - w/o CoT | 1976.12 | 10.03 | 1966.08 | 0.39 | 0.0385 |
| VRPRM-MiMo | 2994.63 | 1028.54 | 1966.08 | 20.40 | 1.8435 |
| - w/o CoT | 1976.12 | 10.03 | 1966.08 | 0.40 | 0.04 |
| VRPRM-Qwen3 | 2410.66 | 584.34 | 1826.31 | 23.06 | 2.0815 |
| - w/o CoT | 1836.35 | 10.03 | 1826.31 | 0.30 | 0.0298 |
| VisualPRM-8B | 1344.88 | 10.03 | 1334.85 | 0.071 | 0.0071 |

## 4.3 BEST-OF-N EVALUATION RESULTS

We use VRPRM as the evaluation model for the BoN task with N set to 8. The InternVL2.5 Chen et al. (2025b) policy model generates N responses through a Chain of Thought (CoT) reasoning process with a temperature of 0.7. The highest-scoring response is selected as the final result. Some results are sourced from the OpenCompass leaderboard Buitrago & Nystrom (2019).

As shown in Table 3, VRPRM significantly improves performance on multiple multimodal reasoning benchmarks. When integrated into the InternVL2.5-8B model, VRPRM led to substantial improvements across all sub-datasets, achieving an overall relative improvement of up to 41.82% over the state-of-the-art VisualPRM. Using VRPRM as a critic model, the InternVL2.5-8B model, with fewer than 10B parameters, outperformed leading proprietary models such as GPT-4o, Claude-3.5, and Gemini-2.0-Flash in reasoning tasks. This demonstrates that test-time scaling can unlock

the latent reasoning potential of foundation models. Similar trends were observed for the larger InternVL2.5-26B and InternVL2.5-38B models.

In summary, the open-source InternVL2.5 model, combined with VRPRM, outperforms proprietary models across multiple tasks using the Best-of-8 strategy, especially in tasks requiring advanced logical reasoning, such as MathVerse-VO and LogicVista. This confirms that VRPRM, trained using our hybrid data method, significantly enhances process evaluation and cross-task transferability in large multimodal models for complex tasks.

Table 3: **Best-of-8 Results on five multimodal reasoning benchmarks.** For MathVerse, we report the performance on Vision-Only (VO) split. The overall score is the average score of the above benchmarks.

| Model | MathVista | MathVision | MathVerse-VO | WeMath | LogicVista | Overall |
|---|---|---|---|---|---|---|
| Proprietary Models | | | | | | |
| GPT-4o | 60.00 | 31.20 | 40.60 | 45.80 | 52.80 | 46.08 |
| Gemini-2.0-Flash | 70.40 | 43.60 | 47.80 | 47.40 | 52.30 | 52.30 |
| Claude-3.5-Sonnet | 65.30 | 35.60 | 46.30 | 44.00 | 60.40 | 50.32 |
| Open-source Models | | | | | | |
| InternVL2.5-8B | 64.50 | 17.00 | 22.80 | 23.50 | 36.38 | 32.84 |
| +VisualPRM | 68.50 | 25.70 | 35.80 | 36.50 | 43.80 | 42.06 |
|  | +4.00 | +8.70 | +13.00 | +13.00 | +7.80 | +9.30 |
| +VRPRM w/o RL | 72.60 | 33.95 | 39.85 | 44.29 | 64.43 | 51.02 |
|  | +8.10 | +16.95 | +17.05 | +20.79 | +28.05 | +18.19 |
| +VRPRM | 79.10 | 51.44 | 51.52 | 36.71 | 79.46 | 59.65 |
|  | **+14.60** | **+34.44** | **+28.72** | **+13.21** | **+43.08** | **+27.23** |
| InternVL2.5-26B | 68.20 | 23.40 | 24.00 | 30.90 | 39.64 | 37.23 |
| +VisualPRM | 73.10 | 29.60 | 39.10 | 40.80 | 51.00 | 46.72 |
|  | +4.9 | +6.20 | +15.10 | +9.90 | +11.40 | +9.50 |
| +VRPRM w/o RL | 77.40 | 37.99 | 44.29 | 48.76 | 68.90 | 55.47 |
|  | +9.20 | +14.59 | +20.29 | +17.86 | +29.26 | +18.24 |
| +VRPRM | 81.20 | 55.79 | 53.55 | 40.14 | 83.00 | 62.74 |
|  | **+13.00** | **+32.39** | **+29.55** | **+9.24** | **+43.36** | **+25.51** |
| InternVL2.5-38B | 71.90 | 32.20 | 36.90 | 38.30 | 47.90 | 45.44 |
| +VisualPRM | 73.90 | 35.20 | 46.70 | 46.20 | 53.70 | 51.14 |
|  | +2.00 | +3.00 | +9.80 | +7.90 | +5.80 | +5.70 |
| +VRPRM w/o RL | 78.40 | 43.45 | 51.52 | 51.43 | 70.02 | 58.96 |
|  | +6.50 | +11.25 | +14.62 | +13.13 | +22.12 | +13.52 |
| +VRPRM | 83.50 | 59.41 | 58.76 | 46.86 | 84.78 | 66.66 |
|  | **+11.60** | **+27.21** | **+21.86** | **+8.56** | **+36.88** | **+21.22** |

## 4.4 ABLATION STUDIES

### 4.4.1 EFFECTS OF BoN

In this experiment, to further verify the cross-model generalization ability of our method, we conducted BoN experiments on four benchmark datasets: LogicVista, MathVerse-VO, MathVista, and MathVision, using the InternVL2.5-8B and Qwen2.5-VL-7B models as policy models. We evaluated the following discriminative models: VRPRM w/o RL, VRPRM, VRPRM-Qwen3, and the baseline models VisualPRM and MM-PRM.

As shown in Figure 4 and Figure 5, the inference accuracy of InternVL2.5-8B and Qwen2.5-VL-7B steadily improved with the increase of the number of candidates N. VRPRM and VRPRM-Qwen3 both showed significant performance improvements, outperforming the MM-PRM baseline model and the majority voting method (Major@K) on all datasets. Notably, the performance of MM-PRM tends to plateau or improve slowly as N increases (e.g., on the LogicVista dataset), while the variant of VRPRM maintains a strong upward trend, significantly narrowing the gap with the Pass@K upper limit. Furthermore, the comparable performance of VRPRM and VRPRM-Qwen3 demonstrates that our training paradigm is effective for different model architectures, including those with intrinsic thinking capabilities.

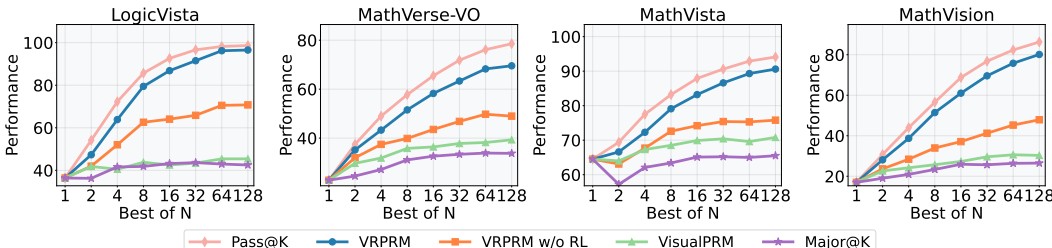

Figure 4: Best-of-N results of InternVL2.5-8B across four multimodel reasoning benchmarks using VisualPRM, VRPRM w/o RL, and VRPRM as critic models. The result of Pass@K is the upper bound.

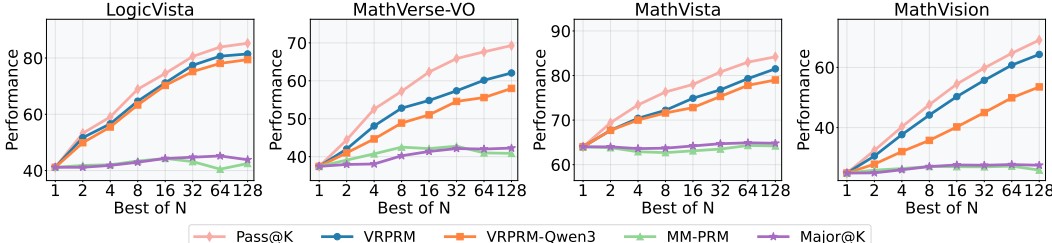

Figure 5: Best-of-N results of Qwen2.5-VL-7B across four multimodel reasoning benchmarks using VRPRM, VRPRM-Qwen3, and MM-PRM as critic models. The result of Pass@K is the upper bound.

### 4.4.2 EFFECTS OF COT

In this experiment, we removed the model's chain of thought reasoning module so that the model no longer performs explicit reasoning when evaluating multi-step solutions. This aims to observe whether VRPRM can effectively utilize CoT reasoning to improve its performance and to analyze the associated computational trade-offs.

**Performance Gain.** The results in Table 1 reveal that removing the CoT module leads to a significant degradation in evaluation performance across all metrics. For instance, for VRPRM-Qwen, the Average All Error Identification (AEI) drops sharply from 66.00 to 53.66, and the First Error Identification (FEI) declines from 49.72 to 38.14. A similar trend is observed in the model trained on MiMo and Qwen3, where the full VRPRM outperforms its non-reasoning counterpart. These results confirm that the model's ability to perform fine-grained error correction is heavily dependent on the intermediate reasoning steps.

**Computational Overhead.** We acknowledge that this performance improvement comes at the cost of increased latency and token consumption. As detailed in Table 2, enabling CoT reasoning increases the average output tokens per sample from 10.03 to 339.73 for VRPRM, resulting in a corresponding increase in processing time from 0.39 to 16.94 seconds per sample. However, this overhead is a necessary trade-off. As illustrated above, the computation allocated to the reasoning process enables the model to perform fine-grained error identification. Also, unlike traditional reward models, VRPRM can provide transparent reasoning processes, transforming from a scorer to a white-box trustworthy verifier. An example output is provided in Appendix G.

In summary, while CoT reasoning introduces computational overhead, it is indispensable for enhancing reward modeling performance. It allows the model to better understand causal relationships and logic between steps, improving its ability to evaluate complex reasoning and make more accurate judgments. Without this capability, the model is more prone to misunderstand intermediate steps, leading to lower evaluation quality.

### 4.4.3 EFFECTS OF RL

In this experiment, we investigated whether reinforcement learning (RL) could improve a model's process evaluation capabilities. The performance of the VRPRM model without RL training (VR-

PRM w/o RL) is reported on the VisualProcessBench and BoN test sets in Tables 1 and 3, respectively.

On the VisualProcessBench, the VRPRM w/o RL, trained with CoT-PRM data during supervised fine-tuning (SFT), outperformed VisualPRM, the state-of-the-art multimodal PRM, in both average FEI and AEI. We then applied RL training to the VRPRM w/o RL using PRM data, creating the complete VRPRM model. This resulted in a 3.92% average performance improvement on VisualProcessBench, with gains across all sub-datasets. In the BoN test, VRPRM consistently outperformed VRPRM without RL across various InternVL model scales, with a maximum relative improvement of 9.04%.

These results show that RL training based on non-CoT PRM data significantly enhances process evaluation capabilities. By incorporating RL, we can effectively train a PRM model with improved evaluation skills at a relatively low data cost.

### 4.4.4 RL TRAINING DYNAMICS

To verify the stability of our reinforcement learning process, we visualize the training trajectories of VRPRM in Figure 6. The curves track the Overall Reward, Format Score, Process Score, and Response Length across training steps. As illustrated, the Overall Reward exhibits a consistent upward trend before converging to a stable plateau, indicating that the model effectively optimizes the objective function via GRPO without experiencing significant crashes or spikes. This confirms the robustness of our RL formulation. Notably, the Response Length begins at a higher value but gradually decreases and stabilizes during training. This suggests that the model learns to generate more efficient and concise reasoning paths rather than exploiting length-based reward hacking. Collectively, these dynamics demonstrate a healthy and stable training process.

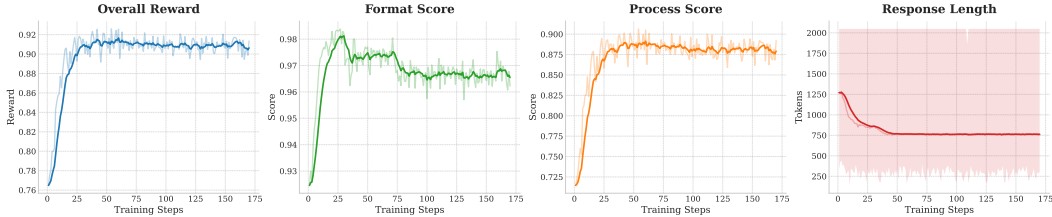

Figure 6: VRPRM training curves. Evolution of reward metrics (Overall, Format, Process) and average response length during the RL fine-tuning stage.

## 5 CONCLUSION

In this paper, we introduce VRPRM, the first Visual Reasoning Process Reward Model capable of incorporating RL reasoning. We have designed a two-stage training strategy for this model. The first stage involves supervised fine-tuning (SFT) on a small set of high-quality CoT data to "activate" the model's reasoning potential. This is followed by a second stage of "reinforcement" through reinforcement learning (RL) using a large volume of lower-cost non-CoT data. Our approach addresses the common deficiency in deep reasoning abilities found in existing process reward models and mitigates the prohibitively high data annotation costs associated with introducing CoT capabilities.

Experimental results demonstrate that VRPRM comprehensively outperforms non-thinking visual process reward models trained on 400K data instances, while using only one-eighth of the training data. This proves the exceptional data efficiency of our method. Furthermore, VRPRM exhibits outstanding test-time scaling capabilities, achieving up to a 118% relative performance improvement on multiple multimodal reasoning benchmarks. This demonstrates that VRPRM is also an effective test-time scaling strategy.

In conclusion, VRPRM offers a novel training paradigm for the future development of process reward models, which can significantly enhance the model's complex reasoning and evaluation capabilities while substantially reducing annotation costs. We believe that this data-efficient training strategy not only carves out a new path for multimodal reward modeling but also provides valuable insights for building more powerful and generalizable reward models in a broader range of fields in the future.

## 6 ETHICS STATEMENT

We acknowledge the ICLR Code of Ethics and affirm that our work complies with its principles. Our research does not raise any immediate ethical concerns. We have considered the potential broader impacts of our work, and we detail our considerations below.

About data usage and privacy, the datasets used in this study are publicly available and were collected in accordance with their original licenses. Our work does not involve the collection of personal data. We have adhered to best practices in data anonymization and privacy preservation where applicable.

About potential biases and fairness, the methods proposed in this work has little risk on introducing biases and unfairness. We have taken steps to mitigate such risks. We encourage further scrutiny and responsible use of our methodology.

About social impact, we believe our research contributes positively. We do not foresee our work being used for malicious purposes, but we acknowledge that any technology can be misused. We encourage the community to use our work responsibly.

About human subjects, this study did not involve human subjects, and no ethical approval was required.

About conflicts of interest: The authors declare no conflicts of interest.

## 7 REPRODUCIBILITY STATEMENT

To facilitate the reproducibility of our work, we have made the following efforts:

About availability, we provide a complete, anonymized implementation of our proposed VRPRM framework, including training and evaluation scripts, as supplementary material. The code will be made publicly available upon publication. The code, models and datasets used in our work are almost all open source and can be easily accessed from the internet.

About experimental setup, our experimental setup is comprehensively documented in the experiment section and appendix to allow for exact replication of our results.

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

## A   ROLLOUT PROMPT AND DATA STATISTICS

In this section we give a Prompt for synthetic data and an example of synthetic data. The prompt for using Claude-3.7-Sonnet to synthetic CoT-PRM Data is shown in Fig 9. The example of CoT-PRM Data is shown in Fig 10.

We report the statistics of CoT-PRM Data. As shown in Fig 7, in CoT-PRM Data, more than 90% of the responses have a thought length of more than 1500 characters, which shows that CoT-PRM Data has good response quality and is a high-quality long-range reasoning process label dataset.

The step distribution statistics of CoT-PRM Data are shown in Fig 8. We observe that most solutions consist of fewer than 15 steps. Among these solutions with fewer than 15 steps, the number of steps has a sample distribution.

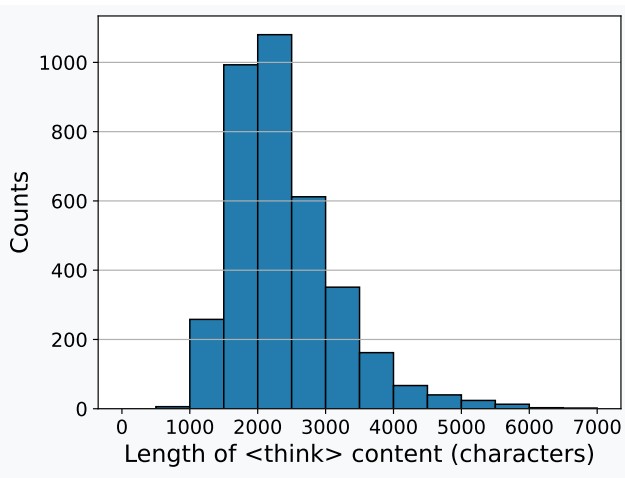

Figure 7: Distribution of think Content Length

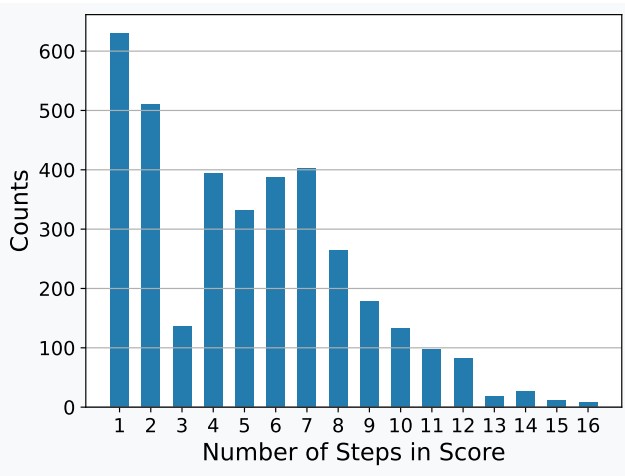

Figure 8: Distribution of Step Count

---

**Prompt for Synthetic CoT-PRM Data**

**[User]:**
You are a reasoning evaluator. Your task is to analyze problem-solving steps one by one. At the same time, according to the analysis process, judge whether the entire problem-solving is correct.

For each solution step, you need to evaluate:
Score (0 or +1):
* +1: Completely correct reasoning
* 0: Completely incorrect
* Use two integers to determine whether the step is correct

For the entire problem-solving, you need to evaluate:
* +1: Completely correct reasoning
* 0: Completely incorrect

Requirements:
- Analysis each step independently and provide scores as integer numbers. After analyzing each step, the analysis results of each step are given in the form of \boxed{Score}
- Evaluate the entire problem-solving and determine whether it is correct
- The scores of the evaluation steps are returned in strict JSON format: "Score": [scores], Ensure arrays have the same length with the number of solution steps
- Consider logical accuracy, mathematical coherence, and solution efficiency

Example output format:
<Step judgment >
Analysis of each step, \boxed{1}
<The score of all steps >
{"Score": [1, 1, 0]}

Question:
{question}
Answer:
{answer}

You will gradually receive each step:

**[User]:**
Step {i}: {the i-th step}

Figure 9: Prompt for Synthetic CoT-PRM Data

## B  MULTIMODAL REASONING BENCHMARKS

We selected five multimodal reasoning benchmarks:

**MathVista Lu et al. (2024)** is a benchmark specifically designed to evaluate the capabilities of Multimodal Large Language Models (MLLMs) in visual mathematical reasoning. The dataset contains 6,141 examples, sourced from 28 existing multimodal math-related datasets, along with three newly created subsets: IQTest, FunctionQA, and PaperQA. MathVista covers a wide range of tasks, including image interpretation, chart reasoning, geometry problem solving, and function graph analysis, aiming to assess models' abilities in processing complex visual information and performing compositional mathematical reasoning. We selected its mini test set, about 1000 samples.

**MathVision Wang et al. (2024)** is a meticulously constructed high-quality benchmark dataset designed to evaluate the visual mathematical reasoning abilities of MLLMs. The dataset contains 3,040

**An example of CoT-PRM Data**

**< think >**

I need to evaluate each step of the problem-solving process and determine if it's correct. Let me analyze each step:

Step 0: This step correctly explains that we need to consider the temperature difference between the liquid and room temperature, and that dew forms on cold surfaces. This is a good setup of the problem.

Step 1: Container A calculation:- Liquid temperature: 36°C - Room temperature: 25°C - Temperature difference: 36°C - 25°C = 11°C. This calculation is mathematically correct.

Step 2: Container B calculation: - Liquid temperature: 63°C - Room temperature: 25°C - Temperature difference: 63°C - 25°C = 38°C. This calculation is mathematically correct.

Step 3: Container C calculation: - Liquid temperature: 83°C - Room temperature: 25°C - Temperature difference: 83°C - 25°C = 58°C. This calculation is mathematically correct.

Step 4: This is where the reasoning error occurs. The step claims that Container A will form dew first because it has the smallest temperature difference. However, this is incorrect. Dew (water droplets) forms on surfaces that are COLDER than the surrounding air, not warmer. All containers have liquids that are warmer than room temperature (25°C).

For dew to form on the inner walls of the containers, the inner wall temperature would need to be below the dew point of the air inside the container. Since all liquids are warmer than room temperature, they will be warming the air inside their containers, not cooling it. When warm, humid air comes in contact with a cold surface, condensation occurs.

Therefore, Container C should show droplets first because it has the highest temperature difference, which creates the greatest potential for condensation when the warm, humid air inside contacts the relatively cooler walls.

Step 5: The final answer A is incorrect based on the faulty reasoning in Step 4.

So: - Steps 0-3: Correct (+1 each) - Step 4: Incorrect (0) - Step 5: Incorrect (0) since it's based on the incorrect reasoning from Step 4

The entire problem-solving process is incorrect because the core reasoning about how condensation forms is flawed.

**< /think >**

**< Step judgment >**

Step 0: This step correctly explains that droplet formation involves temperature differences between the liquid and room temperature. However, there's a fundamental misunderstanding about when condensation occurs. Dew forms when warm, moist air contacts a surface that's cooler than the dew point temperature. \boxed{1}

Step 1: The calculation for Container A is mathematically correct (36°C - 25°C = 11°C). But simply calculating the temperature difference doesn't determine which container will form droplets first. \boxed{1}

Step 2: The calculation for Container B is mathematically correct (63°C - 25°C = 38°C). Again, simply calculating the difference is not incorrect, but the application of this will be important. \boxed{1}

Step 3: The calculation for Container C is mathematically correct (83°C - 25°C = 58°C). The mathematical operation is performed correctly. \boxed{1}

Step 4: This step contains the critical error. The reasoning incorrectly concludes that the container with the smallest temperature difference (Container A) will form dew first. In reality, when hot liquid is placed in a container at room temperature, the warmer liquid creates water vapor. The container with the highest temperature (Container C) will produce the most vapor, which will condense on the relatively cooler inner walls of the container (which are closer to room temperature). The larger the temperature difference, the more pronounced this effect will be. \boxed{0}

Step 5: The final answer A is incorrect as a result of the flawed reasoning in Step 4. Container C (the hottest) would be expected to show condensation first due to the greatest temperature differential producing the most water vapor. \boxed{0}

**< The score of all steps >**

{"Score": [1, 1, 1, 1, 0, 0]}

Figure 10: An Example of CoT-PRM Data

mathematical problems, all sourced from real-world math competitions. It spans 16 distinct mathematical disciplines and is categorized into 5 levels of difficulty, offering a comprehensive assessment across a wide range of topics and complexities. Its complete test set has about 3,000 samples.

**MathVerse Zhang et al. (2024b)** is a comprehensive visual math benchmark designed to provide fair and in-depth evaluation of mathematical diagram understanding and reasoning abilities in MLLMs. The dataset consists of 2,612 high-quality, multi-subject math problems with accompanying diagrams. Each problem is manually transformed into six distinct multimodal versions, varying in the degree of visual and textual information provided, resulting in a total of approximately 15,000 test samples. This design enables MathVerse to rigorously assess whether, and to what extent, MLLMs truly rely on visual diagrams for mathematical reasoning. We report the performance on the Vision-Only split.

**WeMath Qiao et al. (2024)** is the first benchmark specifically designed to explore the underlying problem-solving mechanisms of Multimodal Large Language Models (MLLMs) in visual mathematical reasoning. Rather than focusing solely on final answer accuracy, We-Math emphasizes how models apply knowledge during the reasoning process. The dataset consists of 6,500 carefully curated visual math problems, covering 67 hierarchical knowledge concepts across 5 levels of knowledge granularity, forming a structured and comprehensive knowledge evaluation framework. We report "Score (Strict)" as the main indicator on its mini-test set of about 1740 samples.

**LogicVista Xiao et al. (2024)** is a benchmark specifically designed to evaluate the fundamental logical reasoning abilities of Multimodal Large Language Models (MLLMs) within visual contexts. It focuses on five core categories of logical reasoning tasks: spatial reasoning, deductive reasoning, inductive reasoning, numerical reasoning, and mechanical reasoning, offering a comprehensive assessment across key dimensions of logic.The dataset comprises 448 multiple-choice visual questions drawn from diverse sources and question types, aiming to systematically assess the strengths and limitations of current MLLMs in solving visual logic problems.

## C    MORE RESULTS ON COMPUTATION OVERHEAD

In Table 4, we give detailed token and time results of VRPRM, VRPRM-MiMo, VRPRM-Qwen3 and VisualPRM across VisualProcessBench.

## D    MORE ABLATION RESULTS

In Table 5, we give detailed Best-of-N results on InternVL2.5-8B across four multimodel reasoning benchmarks using VisualPRM, VRPRM w/o RL, and VRPRM as a critic model. The Pass@K results are provided as an upper bound, and the Major@K results are provided as a voting baseline.

## E    CROSS-MODEL GENERALIZATION RESULTS

In Table 6, we give detailed Best-of-N results on Qwen2.5VL-7B across four multimodel reasoning benchmarks using MM-PRM, VRPRM, VRPRM-Qwen3 as a critic model. The Pass@K results are provided as an upper bound, and the Major@K results are provided as a voting baseline.

The results presented in Table 6 demonstrate that our VRPRM training methodology effectively generalizes to different policy models and base architectures, including reasoning model **Qwen3-VL-4B-Thinking** (referred to as VRPRM-Qwen3). Consistent with the findings on the InternVL2.5 policy, the experiments on the Qwen2.5-VL-7B policy show that inference accuracy improves significantly with an increasing number of response candidates $N$, while the performance gap between our VRPRM critics and the baselines also widens, approaching the upper bound Pass@K results.

Taking **LogicVista** as a prime example, both VRPRM and VRPRM-Qwen3 exhibit superior performance. At $N = 128$, VRPRM achieves an accuracy of 81.43, and VRPRM-Qwen3 reaches 79.42. These scores not only substantially outperform the MM-PRM critic (42.51) and the Major@K voting baseline (43.85) by over 35 points but also closely approach the theoretical upper bound of Pass@K (85.23). Similar trends are observed across the other benchmarks, such as MathVerse-VO and MathVision, where our methods consistently dominate the baselines.

| **MMMU Statistics** | | | | |
| Model | Total Tokens / Sample | Output Tokens / Sample | Input Tokens / Sample | Time / Sample (s) | Time / Reward (s) |
| --- | --- | --- | --- | --- | --- |
| VRPRM | 2305.82 | 339.73 | 1966.08 | 16.94 | 1.5353 |
| - w/o CoT | 1976.12 | 10.03 | 1966.08 | 0.39 | 0.0385 |
| VRPRM-MiMo | 2994.63 | 1028.54 | 1966.08 | 20.40 | 1.8435 |
| - w/o CoT | 1976.12 | 10.03 | 1966.08 | 0.40 | 0.04 |
| VRPRM-Qwen3 | 2410.66 | 584.34 | 1826.31 | 23.06 | 2.0815 |
| - w/o CoT | 1836.35 | 10.03 | 1826.31 | 0.30 | 0.0298 |
| VisualPRM-8B | 1344.88 | 10.03 | 1334.85 | 0.071 | 0.0071 |

| **MathVision Statistics** | | | | |
| Model | Total Tokens / Sample | Output Tokens / Sample | Input Tokens / Sample | Time / Sample (s) | Time / Reward (s) |
| --- | --- | --- | --- | --- | --- |
| VRPRM | 2464.82 | 354.12 | 2110.70 | 23.75 | 2.1871 |
| - w/o CoT | 2120.53 | 9.83 | 2110.70 | 0.31 | 0.0314 |
| VRPRM-MiMo | 3587.11 | 1476.41 | 2110.70 | 37.85 | 3.4944 |
| - w/o CoT | 2120.53 | 9.83 | 2110.70 | 0.33 | 0.0336 |
| VRPRM-Qwen3 | 2537.94 | 580.52 | 1957.42 | 21.78 | 2.0082 |
| - w/o CoT | 1967.25 | 9.83 | 1957.42 | 0.23 | 0.0230 |
| VisualPRM-8B | 1480.39 | 9.83 | 1470.56 | 0.0829 | 0.0084 |

| **MathVerse Statistics** | | | | |
| Model | Total Tokens / Sample | Output Tokens / Sample | Input Tokens / Sample | Time / Sample (s) | Time / Reward (s) |
| --- | --- | --- | --- | --- | --- |
| VRPRM | 3466.49 | 333.78 | 3132.71 | 24.44 | 2.3435 |
| - w/o CoT | 3412.14 | 9.42 | 3132.71 | 0.25 | 0.0268 |
| VRPRM-MiMo | 4434.05 | 1301.33 | 3132.71 | 38.22 | 3.6670 |
| - w/o CoT | 3142.14 | 9.42 | 3132.71 | 0.31 | 0.0331 |
| VRPRM-Qwen3 | 3211.13 | 518.60 | 2692.53 | 22.20 | 2.1290 |
| - w/o CoT | 2701.95 | 9.42 | 2692.53 | 0.22 | 0.0229 |
| VisualPRM-8B | 1185.09 | 9.42 | 1175.67 | 0.0646 | 0.0069 |

| **DynaMath Statistics** | | | | |
| Model | Total Tokens / Sample | Output Tokens / Sample | Input Tokens / Sample | Time / Sample (s) | Time / Reward (s) |
| --- | --- | --- | --- | --- | --- |
| VRPRM | 1900.15 | 312.95 | 1587.20 | 22.28 | 2.2683 |
| - w/o CoT | 1596.02 | 8.82 | 1587.20 | 0.21 | 0.0240 |
| VRPRM-MiMo | 2766.17 | 1178.97 | 1587.20 | 27.57 | 2.8012 |
| - w/o CoT | 1596.02 | 8.82 | 1587.20 | 0.27 | 0.0301 |
| VRPRM-Qwen3 | 1975.85 | 506.21 | 1469.63 | 21.22 | 2.1614 |
| - w/o CoT | 1478.45 | 8.82 | 1496.63 | 0.20 | 0.0225 |
| VisualPRM-8B | 2773.59 | 8.82 | 2764.78 | 0.1447 | 0.0164 |

| **Wemath Statistics** | | | | |
| Model | Total Tokens / Sample | Output Tokens / Sample | Input Tokens / Sample | Time / Sample (s) | Time / Reward (s) |
| --- | --- | --- | --- | --- | --- |
| VRPRM | 1782.72 | 334.48 | 1448.24 | 21.77 | 2.2080 |
| - w/o CoT | 1457.10 | 8.86 | 1448.24 | 0.25 | 0.0285 |
| VRPRM-MiMo | 2636.96 | 1188.71 | 1448.24 | 26.12 | 2.6492 |
| - w/o CoT | 1457.10 | 8.86 | 1448.24 | 0.23 | 0.0257 |
| VRPRM-Qwen3 | 1880.29 | 498.62 | 1381.67 | 21.32 | 2.1592 |
| - w/o CoT | 1390.53 | 8.86 | 1381.67 | 0.22 | 0.0246 |
| VisualPRM-8B | 2582.33 | 8.86 | 2573.47 | 0.1352 | 0.0153 |

Table 4: **Computation overhead analysis on VisualProcessBench.** Metrics include average token counts and processing time per sample/reward.

These findings highlight the value of our mixed-data training strategy in building Process Reward Models with greater generalizability and transferability. Furthermore, the strong performance of VRPRM-Qwen3 confirms that this training paradigm is equally effective when extended to reasoning models, enabling them to serve as robust critics even when verifying outputs from different model families.

## F  STRESS TEST ON HUMANITY'S LAST EXAM (HLE)

To evaluate the upper limits and robustness of VRPRM on extremely challenging, out-of-distribution tasks, we conducted a stress test on the *Humanity's Last Exam* (HLE) dataset using the image-enabled subset. For this experiment, we employed a state-of-the-art proprietary model, `gpt-5-mini-2025-08-07`, as the policy model to generate candidate responses. We compared our VRPRM against the baseline reward model VisualPRM-8B and the Self-Consistency baseline (Major@K).

The Best-of-N outcomes are presented in Table 7.

The results on this frontier benchmark provide critical insights into the capabilities of process reward models:

- **Positive Scaling Trend:** As shown in Table 7, the baseline VisualPRM-8B struggles significantly on this dataset, with performance stagnating or even dropping below the Bo1 baseline (10.82% vs 11.11%) as $N$ increases. In stark contrast, VRPRM demonstrates a positive scaling trend, improving from 11.11% to **14.04%** at Bo64.
- **Surpassing Majority Voting:** On extremely difficult tasks where the base accuracy is low ($\approx 11\%$), reward models often fail to outperform the consensus-based Self-Consistency method (Major@K). However, at $N = 64$, VRPRM (**14.04%**) successfully surpasses Major@K (**13.74%**).

This result confirms that VRPRM's process reward signal provides discriminative value beyond simple consensus. Even in scenarios where correct answers are rare ("needles in a haystack"), VRPRM effectively identifies valid reasoning paths, demonstrating strong generalization capability on the most challenging multimodal reasoning scenarios available.

## G  EXAMPLE OF VRPRM OUTPUT

In Fig 11, we present an evaluation of a VRPRM solution to a geographical problem, including chain-of-thought and process evaluation.

## H  EXAMPLE OF VRPRM SCORING

To demonstrate VRPRM's fine-grained verification capability, we provide a case study from the LogicVista test set, policy Qwen2.5-VL-7B, judge model VRPRM. The Problem input is shown in Figure 12, and the evaluation comparison is shown in Table 8.

## I  THE USE OF LLMs

During the writing and editing of this paper, the author(s) utilized Large Language Models (such as ChatGPT) for text refinement to improve the clarity and accuracy of the language. These tools were primarily used for grammar checking, optimizing phrasing, and enhancing readability. All core ideas, the research design, data analysis, and conclusions are the original work of the author(s). The author(s) take full responsibility for the final content of the manuscript and have carefully reviewed all AI-assisted modifications.

**An example of CoT-PRM Data**

[Question]
Which of these states is farthest east?
A. Kentucky
B. Oklahoma
C. Oregon
D. Iowa

[Solution]
**Step 0:**
To determine which state is farthest east, we need to consider their positions relative to the Prime Meridian or the easternmost points of each state.
**Step 1:**
- Oregon is located on the West Coast of the U.S. - Iowa is located in the midwestern United States. - Kentucky is located in the eastern United States. - Oklahoma is located in the central United States.
**Step 2:**
Comparing these positions, Kentucky is clearly the farthest east among them.
**Step 3:**
Answer: \boxed{A}

[Answer]
**<think>**
I'll analyze each step of this problem-solving process to determine if the reasoning is correct.
Step 0: This step correctly establishes the approach - comparing the relative positions of the states on the map to determine which is farthest east. The reasoning is sound. \boxed{1}
Step 1: This step accurately describes the geographic positions of each state as shown in the map. Kentucky is indeed in the east, Iowa and Oklahoma are in the central/midwest, and Oregon is on the west coast. \boxed{1}
Step 2: This step correctly concludes that Kentucky is the farthest east among the four options based on their positions on the map. \boxed{1}
Step 3: The answer selection A (Kentucky) is correct based on the analysis of positions shown in the map. \boxed{1}
The entire problem-solving process is correct. The solver properly analyzed the map, identified the relative positions of all four states, and correctly determined that Kentucky is the farthest east.
**</think>**

**<Step judgment>**
Step 0: This step correctly establishes the approach to compare the positions of the states relative to eastward direction. The methodology is appropriate for solving the problem. \boxed{1}

Step 1: This step accurately describes the geographic positions of each state as shown in the map. The states are correctly categorized based on their location relative to east. \boxed{1}

Step 2: The reasoning correctly identifies Kentucky as the farthest east among the four options based on the map. \boxed{1}

Step 3: The conclusion correctly selects option A (Kentucky) as the answer. \boxed{1}

**<The score of all steps>**
{"Score": [1, 1, 1, 1]}

Figure 11: An Example of VRPRM Output

**Problem Input (Question & Image)**

**Question:** Which set does the Figure belong to? Select from A, B, and C.
(A) Set A
(B) Set B
(C) Neither set A nor set B

Figure 12: Example Input of VRPRM Scoring

| Model | BoN | LogicVista | MathVerse-VO | MathVista | MathVision |
|---|---|---|---|---|---|
| Pass@K | 1 | 36.38 | 22.80 | 64.50 | 17.00 |
| | 2 | 54.14 | 37.44 | 69.40 | 30.76 |
| | 4 | 72.26 | 48.98 | 77.50 | 43.98 |
| | 8 | 85.68 | 57.74 | 83.20 | 56.55 |
| | 16 | 92.62 | 65.48 | 87.90 | 68.75 |
| | 32 | 96.64 | 71.83 | 90.60 | 76.81 |
| | 64 | 98.21 | 76.14 | 92.90 | 82.34 |
| | 128 | 98.66 | 78.55 | 94.10 | 86.28 |
| Major@K | 1 | 36.38 | 22.80 | 64.50 | 17.00 |
| | 2 | 36.24 | 24.49 | 57.20 | 19.01 |
| | 4 | 41.61 | 27.16 | 62.10 | 20.92 |
| | 8 | 41.83 | 31.09 | 63.40 | 23.36 |
| | 16 | 43.18 | 32.61 | 65.10 | 25.92 |
| | 32 | 43.62 | 33.38 | 65.20 | 25.66 |
| | 64 | 42.95 | 33.88 | 65.00 | 26.38 |
| | 128 | 42.51 | 33.76 | 65.50 | 26.48 |
| VisualPRM | 1 | 36.38 | 22.80 | 64.50 | 17.00 |
| | 2 | 41.83 | 29.70 | 64.00 | 22.63 |
| | 4 | 40.49 | 31.85 | 67.30 | 24.18 |
| | 8 | 43.80 | 35.80 | 68.50 | 25.70 |
| | 16 | 42.50 | 36.40 | 69.90 | 27.30 |
| | 32 | 43.40 | 37.80 | 70.40 | 29.60 |
| | 64 | 45.40 | 38.20 | 69.60 | 30.60 |
| | 128 | 45.40 | 39.30 | 70.80 | 30.30 |
| VRPRM w/o RL | 1 | 36.38 | 22.80 | 64.50 | 17.00 |
| | 2 | 41.96 | 31.98 | 63.10 | 23.65 |
| | 4 | 52.01 | 37.44 | 67.70 | 28.42 |
| | 8 | 62.60 | 39.85 | 72.60 | 33.95 |
| | 16 | 64.06 | 43.53 | 74.20 | 37.11 |
| | 32 | 65.85 | 46.83 | 75.40 | 41.25 |
| | 64 | 70.54 | 49.75 | 75.30 | 45.26 |
| | 128 | 70.76 | 48.98 | 75.80 | 47.89 |
| VRPRM | 1 | 36.38 | 22.80 | 64.50 | 17.00 |
| | 2 | 47.32 | 35.15 | 66.60 | 28.09 |
| | 4 | 63.84 | 43.27 | 72.30 | 38.72 |
| | 8 | 79.46 | 51.52 | 79.10 | 51.44 |
| | 16 | 86.83 | 58.25 | 83.20 | 61.02 |
| | 32 | 91.52 | 63.32 | 86.60 | 69.57 |
| | 64 | 96.21 | 68.27 | 89.30 | 75.79 |
| | 128 | 96.54 | 69.54 | 90.60 | 80.13 |
| VRPRM-MiMo | 1 | 36.38 | 22.80 | 64.50 | 17.00 |
| | 2 | 49.44 | 32.49 | 67.70 | 27.34 |
| | 4 | 66.22 | 41.50 | 74.90 | 37.66 |
| | 8 | 77.63 | 50.38 | 81.60 | 49.77 |
| | 16 | 86.35 | 56.60 | 85.60 | 61.48 |
| | 32 | 90.83 | 63.07 | 88.10 | 71.09 |
| | 64 | 94.41 | 67.51 | 91.10 | 77.50 |
| | 128 | 94.63 | 71.07 | 92.60 | 82.47 |

Table 5: Best-of-N results of InternVL2.5-8B across four multimodel reasoning benchmarks using VisualPRM, VRPRM w/o RL, VRPRM, VRPRM-MiMo as critic models. The result of Pass@K is the upper bound, and the result of Major@K provides a baseline of voting.

| Model | BoN | LogicVista | MathVerse-VO | MathVista | MathVision |
|---|---|---|---|---|---|
| Pass@K | 1 | 41.16 | 37.44 | 64.00 | 24.90 |
| | 2 | 53.24 | 44.42 | 69.40 | 32.43 |
| | 4 | 59.06 | 52.54 | 73.40 | 40.26 |
| | 8 | 68.90 | 57.23 | 76.30 | 47.63 |
| | 16 | 74.50 | 62.31 | 78.00 | 54.47 |
| | 32 | 80.54 | 65.86 | 80.80 | 59.80 |
| | 64 | 83.89 | 67.64 | 83.00 | 64.70 |
| | 128 | 85.23 | 69.29 | 84.20 | 69.08 |
| Major@K | 1 | 41.16 | 37.44 | 64.00 | 24.90 |
| | 2 | 41.16 | 37.94 | 64.00 | 24.97 |
| | 4 | 41.83 | 38.07 | 63.60 | 25.99 |
| | 8 | 42.95 | 40.23 | 63.70 | 27.07 |
| | 16 | 44.30 | 41.37 | 64.20 | 27.60 |
| | 32 | 44.74 | 42.13 | 64.70 | 27.50 |
| | 64 | 45.19 | 42.01 | 64.90 | 27.73 |
| | 128 | 43.85 | 42.26 | 64.80 | 27.50 |
| MM-PRM | 1 | 41.16 | 37.44 | 64.00 | 24.90 |
| | 2 | 41.83 | 39.09 | 63.80 | 25.86 |
| | 4 | 42.06 | 40.74 | 62.90 | 26.41 |
| | 8 | 43.40 | 42.51 | 62.70 | 27.07 |
| | 16 | 44.30 | 42.13 | 63.10 | 27.01 |
| | 32 | 43.18 | 42.77 | 63.50 | 26.97 |
| | 64 | 40.49 | 40.99 | 64.30 | 27.20 |
| | 128 | 42.51 | 40.86 | 64.20 | 25.89 |
| VRPRM | 1 | 41.16 | 37.44 | 64.00 | 24.90 |
| | 2 | 51.68 | 42.01 | 67.70 | 30.56 |
| | 4 | 56.60 | 48.10 | 70.40 | 37.66 |
| | 8 | 64.65 | 52.79 | 72.20 | 44.18 |
| | 16 | 71.14 | 54.82 | 74.90 | 50.30 |
| | 32 | 77.40 | 57.36 | 76.80 | 55.72 |
| | 64 | 80.64 | 60.15 | 79.30 | 60.76 |
| | 128 | 81.43 | 62.06 | 81.50 | 64.31 |
| VRPRM-Qwen3 | 1 | 41.16 | 37.44 | 64.00 | 24.90 |
| | 2 | 49.89 | 40.99 | 67.70 | 27.86 |
| | 4 | 55.48 | 44.67 | 70.01 | 32.04 |
| | 8 | 63.31 | 48.86 | 71.60 | 35.76 |
| | 16 | 70.25 | 51.02 | 72.80 | 40.23 |
| | 32 | 75.17 | 54.57 | 75.30 | 45.00 |
| | 64 | 78.08 | 55.58 | 77.80 | 49.90 |
| | 128 | 79.42 | 57.99 | 79.00 | 53.52 |

Table 6: Best-of-N results of Qwen2.5-VL-7B across four multimodel reasoning benchmarks using MM-PRM, VRPRM, VRPRM-Qwen3 as critic models. The result of Pass@K is the upper bound, and the result of Major@K provides a voting baseline.

| Model | Bo1 | Bo2 | Bo4 | Bo8 | Bo16 | Bo32 | Bo64 |
|---|---|---|---|---|---|---|---|
| Pass@K (Oracle) | 11.11 | 14.62 | 16.67 | 23.10 | 26.61 | 31.58 | 34.50 |
| Major@K (Baseline) | 11.11 | 11.11 | 11.40 | 12.28 | 12.87 | 13.45 | 13.74 |
| VisualPRM-8B | 11.11 | 10.53 | 10.23 | 10.53 | 10.82 | 10.82 | 10.82 |
| **VRPRM (Ours)** | **11.11** | **11.40** | **11.99** | **11.11** | **11.40** | **13.45** | **14.04** |

Table 7: **Best-of-N Performance on Humanity's Last Exam (HLE).** Comparison of VRPRM against baselines using `gpt-5-mini-2025-08-07` as the policy model. Pass@K represents the theoretical upper bound (Oracle), while Major@K represents Majority Voting.

| Candidate 1: Winner (Correct) | Candidate 2: Loser (Incorrect) |
|---|---|
| **1. Policy Generated Step List** | |

| Candidate 1: Winner (Correct) | Candidate 2: Loser (Incorrect) |
|---|---|
| **Step 1:** To determine which set the figure belongs to, let's analyze the characteristics of the flowers in both sets: 
 **Step 2:** Set A: All flowers have petals that are black on the outside and white on the inside. Set B: All flowers have petals that are white on the outside and black on the inside. 
 **Step 3:** The figure in question has petals that are white on the outside and black on the inside, which matches the pattern of the flowers in Set B. 
 **Step 4:** Therefore, the figure belongs to Set B. 
 **Step 5:** Answer: $\boxed{B}$ | **Step 1:** To determine which set the figure belongs to, let's analyze the characteristics of the flowers in both sets: 
 **Step 2:** **Set A**: All flowers have petals that are black on the outside and white on the inside. **Set B**: All flowers have petals that are white on the outside and black on the inside. 
 **Step 3:** The figure in question has petals that are black on the outside and white on the inside, which matches the pattern of the flowers in Set A. 
 **Step 4:** Therefore, the figure belongs to Set A. 
 **Step 5:** Answer: $\boxed{A}$ |

**2. VRPRM Generated Judgments (CoT)**

| Candidate 1: Winner (Correct) | Candidate 2: Loser (Incorrect) |
|---|---|
| **< think >** 
 Let me analyze this problem-solving process step by step: 

 Step 0: This step correctly identifies the task - analyzing the characteristics of flowers to determine which set the figure belongs to. This is a proper setup for the problem. 

 Step 1: This step accurately describes the characteristics of both sets A and B as shown in the image. Set A has flowers with black outer petals and white inner petals, while Set B has flowers with white outer petals and black inner petals. This is factually correct based on the image. 

 Step 2: This step correctly identifies that the figure in question has petals that are white on the outside and black on the inside, matching Set B's description. 

 Step 3: This step correctly concludes that the figure belongs to Set B based on the previous analysis. 

 Step 4: This step correctly provides the answer as B, which matches the reasoning in the previous steps. 

 All steps are logically sound and lead to the correct conclusion that the figure belongs to Set B (option B). 
 **</think >** 

 **[Step judgment]** 
 **Step 0:** To determine which set the figure belongs to, let's analyze the characteristics of the flowers in both sets: $\boxed{1}$ 
 **Step 1: Set A**: All flowers have petals that are black on the outside and white on the inside. **Set B:** All flowers have petals that are white on the outside and black on the inside. $\boxed{1}$ 
 **Step 2:** The figure in question has petals that are white on the outside and black on the inside, which matches the pattern of the flowers in Set B. $\boxed{1}$ 
 **Step 3:** Therefore, the figure belongs to Set B. $\boxed{1}$ 
 **Step 4:** Answer: $\boxed{B}$ $\boxed{1}$ | **< think >** 
 I need to analyze each step of the problem-solving process and determine if the entire problem-solving is correct. 

 Step 0: This step correctly identifies the task - to analyze the flowers in both sets to determine which set the figure belongs to. This is a good setup for the problem. $\boxed{1}$ 

 Step 1: This step correctly describes the characteristics of both sets A and B as shown in the image. Set A has flowers with black outside and white inside, while Set B has flowers with white outside and black inside. This is accurate based on the image. $\boxed{1}$ 

 Step 2: This step correctly identifies that the figure has petals that are black on the outside and white on the inside, matching Set A. However, this is incorrect because the figure actually matches Set B's description. $\boxed{0}$ 

 Step 3: This step concludes the figure belongs to Set A, which is incorrect based on the image. The figure matches Set B's description. $\boxed{0}$ 

 Step 4: The answer given is A, which is incorrect. Based on the image, the figure belongs to Set B, which corresponds to answer B. $\boxed{0}$ 

 Overall, the problem-solving is incorrect because steps 2, 3, and 4 all misidentify the figure's characteristics and incorrectly conclude it belongs to Set A when it actually belongs to Set B. 
 **</think >** 

 **[Step judgment]** 
 **Step 0:** To determine which set the figure belongs to, let's analyze the characteristics of the flowers in both sets: $\boxed{1}$ 
 **Step 1:** - **Set A**: All flowers have petals that are black on the outside and white on the inside. 
 Set B: All flowers have petals that are white on the outside and black on the inside. $\boxed{1}$ 
 **Step 2:** The figure in question has petals that are black on the outside and white on the inside, which matches the pattern of the flowers in Set A. $\boxed{0}$ 
 **Step 3:** Therefore, the figure belongs to Set A. $\boxed{0}$ 
 **Step 4:** Answer: $\boxed{A}$ $\boxed{0}$ |

**3. VRPRM Quantified Step Rewards ($P_\theta(1|\textbf{context})$)**

| Candidate 1: Winner (Correct) | | Candidate 2: Loser (Incorrect) | |
|---|---|---|---|
| Step 1: | 1.0 | Step 1: | 1.0 |
| Step 2: | 1.0 | Step 2: | 1.0 |
| Step 3: | 1.0 | **Step 3:** | **5.22e-05** ↓ (Error Detected) |
| Step 4: | 1.0 | Step 4: | 1.81e-07 |
| Step 5: | 1.0 | Step 5: | 3.29e-06 |
| **Avg:** | **1.0** (Preferred) | **Avg:** | **0.40** (Not Preferred) |

Table 8: A fully worked example of VRPRM scoring on a visual logic task. The table compares the evaluation of a correct response (Left) with the evaluation of an incorrect one. Row 3 explicitly shows the step-level probabilities ($P(\text{Token} = 1)$). Note VRPRM maintains a score of 1.0 for the correct response but drastically drops the score to near-zero at Step 3 of the incorrect response, effectively identifying the hallucination regarding the flower's petal color.

