# OpenReview forum: "VRPRM: Process Reward Modeling via Visual Reasoning"
_ICLR.cc/2026/Conference — Submitted to ICLR 2026_

### Official Review · Reviewer_jxEj · 2025-10-30

**Soundness:** 3
**Presentation:** 3
**Contribution:** 2
**Rating:** 4
**Confidence:** 4

**Summary:**

The paper proposes VRPRM, a multimodal Process Reward Model (PRM) with explicit Chain-of-Thought (CoT) capabilities, trained via a data-efficient, two-stage paradigm: SFT on a small, high-quality CoT dataset to elicit reasoning, followed by RL amplification on large-scale non-CoT data. Used as a critic for Best-of-N selection, VRPRM significantly improves multimodal reasoning benchmarks and outperforms the SOTA (e.g., VisualPRM) on VisualProcessBench (FEI/AEI) with only ~13.4% of the training data.

**Strengths:**

The method achieves modest performance improvements with better data efficiency

**Weaknesses:**

The techniques are well-explored in the text-only domain; neither CoT-PRM nor RL is novel. The paper also omits discussion of relevant related work. For instance, R-RPM[1] discussed a CoT-based PRM with further optimization using DPO, achieving high data efficiency, and DeepSeek-GRM[2] explored a generative universal reward model in an RL context. Overall, the paper's novelty is insufficient.

The improvement on recent models like MiMo appears marginal. This raises questions about the method's effectiveness, especially given that "thinking" capabilities are now prevalent in models such as Qwen3-4B-VL-Thinking.

[1] R-PRM: Reasoning-Driven Process Reward Modeling
[2] Inference-Time Scaling for Generalist Reward Modeling

**Questions:**

I suggest reporting Major@8 and Pass@8 in Table 2 to better clarify the magnitude of the improvement and its position relative to the upper bound. I also recommend reporting the performance of the teacher model used for distillation.

---

> ### Author Response · Authors · 2025-11-27
> **Response Part 1**
>
> Dear Reviewer jxEj,
>
> We thank the reviewer for acknowledging that "The method achieves modest performance improvements with better data efficiency." Your constructive comments and suggestions are invaluable to us. Below, we provide detailed responses to your questions and concerns.
>
> # Response to Weakness 1: Novelty and Related Work
>
> We appreciate the reviewer highlighting these relevant works. We will discuss and cite **R-PRM** and **DeepSeek-GRM** in the revised "Related Work" section. However, we respectfully argue that the novelty of VRPRM extends far beyond simply adapting text-based techniques to the multimodal domain. Our contribution addresses specific bottlenecks in multimodal reasoning that text-only methods cannot resolve.
>
> While the reviewer correctly notes that these techniques are well-explored in the text domain, applying them to Multimodal Large Language Models (MLLMs) presents unique challenges. In text, reasoning relies solely on semantic coherence. In multimodal tasks, models must ground their reasoning steps in specific visual features. Text-based PRMs lack the capability to verify whether a reasoning step accurately aligns with the image content.
>
> VRPRM is the first Visual PRM to **explicitly integrate CoT reasoning with Reinforcement Learning (RL)**, specifically tailored to address the challenges of visual reasoning. We demonstrate that RL is critical for enforcing the alignment between visual perception and logical reasoning—a capability that standard Supervised Fine-Tuning (SFT) often struggles to acquire robustly. Unlike text-based PRMs, our focus is on **extreme data efficiency** in the multimodal domain, where the cost of collecting Multimodal CoT data is significantly higher than that of text data.
>
> Our key innovation lies in demonstrating that massive multimodal CoT datasets are not strictly necessary. By employing a strategy of **"Cold Start" (3.6K) + "RL Scaling" (50K non-CoT data)**, our model outperforms baselines trained on **400K samples**. This solution to the multimodal data scarcity bottleneck represents a novel contribution that is fundamentally distinct from general text-based GRMs.
>
> In conclusion, while neither CoT-PRM nor RL is novel in isolation, their specific adaptation to solve the **data bottleneck in visual reasoning** is a novel contribution.

---

> > ### Author Response · Authors · 2025-11-27
> > **Response Part 3**
> >
> > # Response to Weakness 2: Effectiveness on Thinking Models
> >
> > We appreciate the reviewer's candid assessment. We understand the concern regarding the marginal performance gains on MiMo and the relevance of PRMs in the era of "thinking" models. To clarify the value of our approach, we offer the following points:
> >
> > 1. **Improvement on Strong Baselines:** As shown in **Table 1**, the baseline **MiMo-VL-7B** exhibits exceptional performance with an average **AEI of 63.38**, significantly surpassing Qwen2.5-VL's 47.82. Achieving further improvements on such a strong baseline is inherently more challenging due to diminishing returns, especially under comparable training data scales. Nevertheless, **VRPRM-7B-MiMo** successfully breaks this bottleneck, improving AEI by **3.06 percentage points** (63.38 $\rightarrow$ 66.44). In the context of reward modeling, identifying subtle logical errors is notoriously difficult; we believe this consistent improvement confirms that our RL strategy remains effective even on base models already equipped with "thinking" capabilities.
> >
> > 2. **The Necessity of PRMs for "Thinking" Models:** The reviewer questions the necessity of VRPRM given the rise of models with intrinsic "thinking" capabilities (e.g., the Qwen3-4B-VL-Thinking series). We argue that for these models, **the importance of PRMs increases rather than decreases.** While "thinking" models are excellent generators, they remain prone to hallucinations or subtle logical deviations over long reasoning spans. VRPRM serves as a critical step-aware verifier. As demonstrated in Table 2, even with a strong policy model, using VRPRM to select the best response from $N$ candidates yields significant performance gains.
> >
> > 3. **Effectiveness on Thinking Models:** We highly value the reviewer's concern regarding our method's effectiveness on thinking models. The training paradigm proposed in this paper is equally applicable to current "thinking" models. To verify, we trained **VRPRM-Qwen3** using **Qwen3-4B-VL-Thinking** (a model with intrinsic thinking capabilities) as the base model using our two-stage method.
> >
> >     - ** Evaluation on VisualProcessBench:** We first evaluated VRPRM-Qwen3 on VisualProcessBench (VPB). As shown in Table 1, VRPRM-Qwen3 achieved the highest AEI (0.6774) and FEI (0.5506), significantly outperforming the VisualPRM-8B baseline (AEI 0.5901) and Qwen2.5-VL-7B itself (AEI 0.4688). This demonstrates that even for models with inherent thinking abilities, our PRM training significantly enhances their capacity to identify logical errors.
> >
> >     - Validation on Best-of-N (BoN): We conducted BoN testing on the Qwen2.5-VL-7B policy model, comparing VRPRM-Qwen3 against VRPRM-Qwen2.5, MM-PRM, as well as Pass@k and Major@k benchmarks. Taking the Bo8 results on LogicVista as an example (see Table 2), VRPRM-Qwen3 achieved 63.31%, far surpassing the MM-PRM baseline (43.40%) and Major@8 (42.95%). It performs comparably to the specialized VRPRM-Qwen2.5 (64.65%) and approaches the upper bound represented by Pass@8. We observed identical trends and performance characteristics across other datasets including MathVerse, MathVista, and MathVision (see Tables 3-5).
> >
> > Conclusion: These results prove that our reward modeling approach is highly compatible with emerging "thinking models," effectively guiding them to select optimal answers and providing robust verification.
> >
> > # Response to Q3: Adding `Pass@k` and `Major@k` Metrics
> >
> > We sincerely thank the reviewer for this valuable suggestion. We agree with the recommendation and have reported the `Major@8` and `Pass@8` results in **Tables 2-5**.

---

> ### Author Response · Authors · 2025-11-27
> **Response Part 2**
>
> | Model                      | AEI    | FEI    | AVG    |
> |----------------------------|--------|--------|--------|
> | VRPRM-Qwen2.5              | 0.6316 | 0.5206 | 0.5761 |
> | - w/o CoT                  | 0.5266 | 0.4644 | 0.4955 |
> | - w/o RL                   | 0.5594 | 0.4757 | 0.5176 |
> | VRPRM-MiMo                 | 0.6484 | 0.5394 | 0.5939 |
> | - w/o CoT                  | 0.6488 | 0.5431 | 0.5960 |
> | - w/o RL                   | 0.5763 | 0.5026 | 0.5395 |
> | VRPRM-Qwen3-4B-VL-Thinking | 0.6774 | 0.5506 | 0.614  |
> | w/o CoT                    | 0.6473 | 0.4794 | 0.5633 |
> | w/o RL                     | 0.5867 | 0.4419 | 0.5143 |
> | VisualPRM-8B               | 0.5901 | 0.3071 | 0.4486 |
> | Qwen2.5VL-7B               | 0.4688 | 0.4457 | 0.4573 |
> Table 1: AEI and FEI Results on VisualProcessBench (MMMU)
>
> | Model            | Bo1   | Bo2   | Bo4   | Bo8   | Bo16  | Bo32  | Bo64  | Bo128 |
> | :--------------- | :---: | :---: | :---: | :---: | :---: | :---: | :---: | :---: |
> | **VRPRM-Qwen2.5** | **41.16** | **51.68** | **56.60** | **64.65** | **71.14** | **77.40** | **80.64** | **81.43** |
> | VRPRM-Qwen3-4B   | 41.16 | 49.89 | 55.48 | 63.31 | 70.25 | 75.17 | 78.08 | 79.42 |
> | MM-PRM           | 41.16 | 41.83 | 42.06 | 43.40 | 44.30 | 43.18 | 40.49 | 42.51 |
> | **Metric (K)**   | **1** | **2** | **4** | **8** | **16**| **32**| **64**| **128**|
> | Major@K          | 41.16 | 41.16 | 41.83 | 42.95 | 44.30 | 44.74 | 45.19 | 43.85 |
> | Pass@K           | 41.16 | 53.24 | 59.06 | 68.90 | 74.50 | 80.54 | 83.89 | 85.23 |
> Table 2: Best-of-N Results on Qwen2.5-VL-7B Policy, LogicVista
>
> | Model            | Bo1   | Bo2   | Bo4   | Bo8   | Bo16  | Bo32  | Bo64  | Bo128 |
> | :--------------- | :---: | :---: | :---: | :---: | :---: | :---: | :---: | :---: |
> | **VRPRM-Qwen2.5** | **37.44** | **42.01** | **48.10** | **52.79** | **54.82** | **57.36** | **60.15** | **62.06** |
> | VRPRM-Qwen3-4B   | 37.44 | 40.99 | 44.67 | 48.86 | 51.02 | 54.57 | 55.58 | 57.99 |
> | MM-PRM           | 37.44 | 39.09 | 40.74 | 42.51 | 42.13 | 42.77 | 40.99 | 40.86 |
> | **Metric (K)**   | **1** | **2** | **4** | **8** | **16**| **32**| **64**| **128**|
> | Major@K          | 37.44 | 37.94 | 38.07 | 40.23 | 41.37 | 42.13 | 42.01 | 42.26 |
> | Pass@K           | 37.44 | 44.42 | 52.54 | 57.23 | 62.31 | 65.86 | 67.64 | 69.29 |
> Table 3: Best-of-N Results on Qwen2.5-VL-7B Policy, MathVerse-VO
>
> | Model            | Bo1   | Bo2   | Bo4   | Bo8   | Bo16  | Bo32  | Bo64  | Bo128 |
> | :--------------- | :---: | :---: | :---: | :---: | :---: | :---: | :---: | :---: |
> | **VRPRM-Qwen2.5** | **64.00** | **67.70** | **70.40** | **72.20** | **74.90** | **76.80** | **79.30** | **81.50** |
> | VRPRM-Qwen3-4B   | 64.00 | 67.00 | 70.10 | 71.60 | 72.80 | 75.30 | 77.80 | 79.00 |
> | MM-PRM           | 64.00 | 63.80 | 62.90 | 62.70 | 63.10 | 63.50 | 64.30 | 64.20 |
> | **Metric (K)**   | **1** | **2** | **4** | **8** | **16**| **32**| **64**| **128**|
> | Major@K          | 64.00 | 64.00 | 63.60 | 63.70 | 64.20 | 64.70 | 64.90 | 64.80 |
> | Pass@K           | 64.00 | 69.40 | 73.40 | 76.30 | 78.00 | 80.80 | 83.00 | 84.20 |
> Table 4: Best-of-N Results on Qwen2.5-VL-7B Policy, MathVista
>
> | Model            | Bo1   | Bo2   | Bo4   | Bo8   | Bo16  | Bo32  | Bo64  | Bo128 |
> | :--------------- | :---: | :---: | :---: | :---: | :---: | :---: | :---: | :---: |
> | **VRPRM-Qwen2.5** | **24.90** | **30.56** | **37.66** | **44.18** | **50.30** | **55.72** | **60.76** | **64.31** |
> | VRPRM-Qwen3-4B   | 24.90 | 27.86 | 32.04 | 35.76 | 40.23 | 45.00 | 49.90 | 53.52 |
> | MM-PRM           | 24.90 | 25.86 | 26.41 | 27.07 | 27.01 | 26.97 | 27.20 | 25.89 |
> | **Metric (K)**   | **1** | **2** | **4** | **8** | **16**| **32**| **64**| **128**|
> | Major@K          | 24.90 | 24.97 | 25.99 | 27.07 | 27.60 | 27.50 | 27.73 | 27.50 |
> | Pass@K           | 24.90 | 32.43 | 40.26 | 47.63 | 54.47 | 59.80 | 64.70 | 69.08 |
> Table 5: Best-of-N Results on Qwen2.5-VL-7B Policy, MathVision

---

### Official Review · Reviewer_SHdC · 2025-10-31

**Soundness:** 3
**Presentation:** 3
**Contribution:** 2
**Rating:** 4
**Confidence:** 3

**Summary:**

This paper introduces VRPRM, a multimodal process reward model that inserts explicit CoT reasoning into PRM training and then scales it with reinforcement learning. The authors motivate PRMs as step-level evaluators used for RL and test-time scaling, noting that existing visual PRMs lack deep reasoning; they position VRPRM as a two-stage alternative: supervised fine-tuning on a small CoT-PRM set followed by RL on larger non-CoT PRM data. In the abstract they claim strong data efficiency, 3.6K CoT-PRM SFT + 50K non-CoT RL surpasses a 400K non-thinking PRM and yields up to 118% relative gains over a base model under Best-of-N selection.

**Strengths:**

- **Simple two-stage recipe combining CoT and RL for easy adoption.** The paper first uses SFT on a small, structured CoT-PRM dataset to seed reasoning, then applies RL on larger non-CoT PRM data to scale. CoT and RL are explicitly integrated in a multimodal PRM, and rewards cover both format and process, keeping implementation straightforward.
- **Stronger process supervision with clear empirical gains.** On *VisualProcessBench* it surpasses prior PRMs on process metrics. ablations show removing CoT degrades performance while adding RL yields further gains, supporting the training recipe.
- **Practical test-time scaling.** Used as a BoN critic at inference, it consistently improves results across multiple multimodal benchmarks and model sizes.

**Weaknesses:**

- My main concern is the inconsistency between the experimental settings of Table 1 and Table 2. In Table 1, the authors evaluate VRPRM using Qwen and MiMo backbones to analyze the model’s process reasoning ability, but in Table 2, they switch to the InternVL2.5 family as the policy model for BoN testing without specifying which version of VRPRM serves as the critic, and they omit the MiMo results entirely. This inconsistency makes the two tables difficult to compare and raises questions about the robustness of the approach across backbones. In addition, the inclusion of results from external leaderboards further weakens the fairness and reproducibility of the reported improvements.
- Another concern is the narrow scope of the evaluation datasets.
  Most benchmarks used in the paper (such as VisualProcessBench, MathVista, MathVision, MathVerse-VO, WeMath, and LogicVista) are all visual–mathematical or logic reasoning datasets.
  While these settings highlight the model’s performance on step-by-step reasoning tasks, they do not demonstrate its generality to broader multimodal understanding domains (e.g., commonsense reasoning, visual QA, or instruction following).
  As a result, the claimed contribution of a “general multimodal PRM” currently lacks sufficient empirical support beyond the math reasoning domain.
- Please clarify what is methodologically new beyond combining CoT-SFT with RL under process/format rewards. In particular, position VRPRM against ATHENA[1] and MM-PRM[2]: what is distinct in your reward signal/design, two-stage training protocol, or test-time critic use? A brief comparison (and, ideally, head-to-head results under a unified pipeline) would resolve concerns that the contribution is primarily an engineering recipe.

[1]  ATHENA: ENHANCING MULTIMODAL REASONING WITH DATA-EFFICIENT PROCESS REWARD MODELS

[2] MM-PRM: Enhancing Multimodal Mathematical Reasoning with Scalable Step-Level Supervision

**Questions:**

Please address the identified weaknesses in your response, and correcting the noted typos and inconsistencies may further improve the paper’s clarity and readability.

- PROMBLEM FORMULATION → PROBLEM FORMULATION.

- The bold and underline annotations in Table 1 appear inconsistent with the numerical values (e.g., some smaller numbers are highlighted while larger ones are not).  Please double-check the highlighting to ensure that the bold font truly marks the best result and the underline the second best.

- The paper introduces many abbreviations (e.g., RM/ORM/PRM/CoT, SFT/RL, BoN, FEI/AEI, MMMU, VO, etc.) and not all are defined at first mention or used consistently across sections/tables.  Please define every acronym upon first use in the main text.

- In the Introduction, Figure 1 is mistakenly referred to as Table 1.

---

> ### Author Response · Authors · 2025-11-27
> **Response Part 1**
>
> Dear Reviewer SHdC,
>
> We sincerely thank you for your positive feedback: "Simple two-stage recipe combining CoT and RL for easy adoption," "Stronger process supervision with clear empirical gains," and "Practical test-time scaling." We address your questions and concerns below.
>
> # Response to W1: Inconsistency Between the Experimental Settings of Table 1 and Table 2
> We sincerely apologize for any confusion caused by the presentation of our experimental settings. We fully agree with the reviewer that consistency is crucial. However, we would like to clarify that the experimental setups in Table 1 and Table 2 follow the experimental setups of VisualPRM and serve distinct objectives:
> 1. **Distinct Roles:**
> In Table 1, Qwen and MiMo serve as the backbones for the PRM (**the Judge**). The objective here is to evaluate the intrinsic performance of the Process Reward Model itself using metrics such as FEI and AEI. We reported results on both backbones to demonstrate that our training strategy is effective across different architectures.
> In Table 2, InternVL2.5 serves as the Policy Model (**the Generator**), while VRPRM acts as the Critic. The objective here is to verify whether our PRM can enhance the performance of existing state-of-the-art multimodal LLMs through Best-of-N (BoN) sampling.
> 2. **Clarification on the Critic Model:**
> For the results in Table 2, we employed VRPRM-7B-Qwen as the critic (we will explicitly state this in the revised Table 2 caption). We selected this variant due to its excellent balance between performance and inference efficiency.
> Furthermore, to address your concern regarding the omission of MiMo, we have completed the BoN experiments using VRPRM-MiMo as the critic with InternVL2.5 as the policy model. The results are presented in Table 6 below and have been included in Table 5 of the revised paper.
> 3. **External Baselines:**
> The inclusion of OpenCompass results (e.g., GPT-4o) is intended solely as a high-level reference to contextualize our method's performance against proprietary SOTA models. This is a standard practice in the community to benchmark the capabilities of open-source models against closed-source counterparts.
>
> # Response to W2: Clarification of the Scope of Evaluation Benchmarks
>
> We thank the reviewer for emphasizing the importance of broader multimodal understanding. However, we wish to clarify a potential oversight regarding our experimental setup: as demonstrated in Table 1, we have indeed evaluated our model on the MMMU benchmark within VisualProcessBench.
>
> We believe the inclusion of MMMU directly addresses the reviewer's concern, as MMMU (Massive Multi-discipline Multimodal Understanding) is specifically designed to evaluate broad, expert-level multimodal capabilities. **Table 1** demonstrates that VRPRM achieves superior performance on this general-purpose benchmark. Specifically, **VRPRM-7B-MiMo** achieves an **FEI of 53.94** and an **AEI of 64.84** on the MMMU test set, significantly surpassing the **VisualPRM-8B** baseline scores of 30.71 and 59.01. Consequently, the performance gains on the MMMU benchmark provide compelling evidence that VRPRM's reasoning capabilities effectively transfer to diverse domains such as medical diagnosis, art analysis, and social science reasoning.
>
> | Model                      | AEI   | FEI   | AVG   |
> |----------------------------|-------|-------|-------|
> | VRPRM-Qwen2.5              | 63.16 | 52.06 | 57.61 |
> | - w/o CoT                  | 52.66 | 46.44 | 49.55 |
> | - w/o RL                   | 55.94 | 47.57 | 51.76 |
> | VRPRM-MiMo                 | 64.84 | 53.94 | 59.39 |
> | - w/o CoT                  | 64.88 | 54.31 | 59.60 |
> | - w/o RL                   | 57.63 | 50.26 | 53.95 |
> | VRPRM-Qwen3-4B-VL-Thinking | 67.74 | 55.06 | 61.40 |
> | w/o CoT                    | 64.73 | 47.94 | 56.33 |
> | w/o RL                     | 58.67 | 44.19 | 51.43 |
> | VisualPRM-8B               | 59.01 | 30.71 | 44.86 |
> | Qwen2.5VL-7B               | 46.88 | 44.57 | 45.73 |
>
> Table 1: AEI and FEI Results on VisualProcessBench (MMMU)

---

> ### Author Response · Authors · 2025-11-27
> **Response Part 2**
>
> # Response to W3: Clarification of Novelty && Comparison with ATHENA and MM-PRM
>
> We sincerely thank the reviewer for highlighting these relevant works. We clarify that VRPRM differs fundamentally from ATHENA and MM-PRM in terms of **model paradigm** (Generative vs. Discriminative), **training objectives**, and **reward signal design**. While ATHENA and MM-PRM focus on **data engineering** (filtering/MCTS) to train discriminative PRMs, VRPRM proposes a **methodological shift**, treating reward modeling as a reasoning task optimization problem.
>
> 1. **Methodological Distinction: Generative CoT-PRM vs. Discriminative PRM**
> Both ATHENA and MM-PRM follow the traditional PRM paradigm where the model acts as a classifier. They take solution steps as input and directly output a scalar score (or probability). ATHENA focuses on removing label noise via consistency filtering between weak/strong completers, while MM-PRM focuses on scaling supervision via soft labels derived from MCTS.
> - **Limitation:** They provide scores without explanation, limiting interpretability and generalization on complex datasets.
> - **VRPRM Advantage:** We redesign the PRM as a "Thinker-Critic". Before assigning a score, VRPRM explicitly generates a Chain-of-Thought (CoT) to reason about the task and problem, scoring the steps based on its own understanding.
> - **Innovation:** This is not merely combining CoT-SFT with Reinforcement Learning (RL). We treat the PRM itself as a policy and use RL to optimize its internal reasoning process for evaluation, rather than just optimizing the final score prediction. This aligns the PRM's evaluation logic with the complexity of the task.
>
> 2. **Training Protocol Distinction: RL Optimization vs. Supervised Fine-Tuning**
> Both ATHENA and MM-PRM employ Supervised Fine-Tuning (SFT). ATHENA uses cross-entropy loss on filtered binary labels, and MM-PRM uses cross-entropy loss on soft labels derived from MCTS. Essentially, they "fit" the pre-collected data distribution.
> VRPRM takes a different approach:
> - We adopt a **"Cold Start + RL"** paradigm: first using SFT to enable the model to produce structured outputs, followed by RL to explore and optimize the PRM's reasoning capability.
> - We use a **hybrid reward signal** in our RL stage: Unlike traditional discriminative PRM training, our reward signals consist of *Format Reward* (to enforce structured thinking) + *Process Accuracy Reward* (to align reasoning with ground truth). This enables the model to **self-correct** and improve its judgment capability beyond the SFT baseline, which is an improvement unattainable by SFT-only methods like ATHENA or MM-PRM.
>
> To demonstrate that our methodology has a fundamental paradigm shift and yields superior performance over the standard Discriminative PRM paradigm (as used in MM-PRM), we conducted a controlled head-to-head evaluation in a BoN experiment under the reviewer's valuable advice. We utilized **Qwen2.5-VL-7B** as the fixed policy model across four diverse benchmarks and compared the performance of **MM-PRM** against our **VRPRM**, as well as the **pass@k** and **major@k** results of the policy model. Results are shown in Table 2-5.
>
> **Observation:** When applied to the Qwen2.5-VL-7B policy, our VRPRM models **consistently and significantly** outperform the MM-PRM baseline across all datasets. MM-PRM struggles in these scenarios because it maps complex visual-textual inputs directly to a scalar score without intermediate reasoning steps.  In contrast, VRPRM's explicit CoT generation allows it to "think through" the logic before scoring. This confirms that our contribution is not merely an engineering recipe, but a fundamental upgrade in how reward models should handle complex multimodal reasoning.
>
> # Response to Typos and Presentation Details
>
> We sincerely thank the reviewer for their **meticulous reading and valuable attention to detail**. We apologize for the oversight and have carefully revised the manuscript to address all the presentation issues raised:
>
> **Typos:** We have corrected the section title from "PROMBLEM FORMULATION" to "PROBLEM FORMULATION."
>
> **Table 1 Formatting:** We have rigorously re-verified all numerical values in Table 1. In this recheck, we identified a data version mismatch, the previously reported data were inadvertently sourced from an earlier checkpoint. We corrected the data and highlighting to ensure that bold font strictly marks the best result and underline marks the second-best result across all columns.
>
> **Abbreviations:** We have conducted a full pass over the manuscript to define all acronyms (e.g., RM, ORM, PRM, CoT, SFT, RL, BoN, FEI, AEI, MMMU, VO) upon their first mention and ensured consistent usage throughout the text.
>
> **Cross-referencing:** We have corrected the error in the Introduction, ensuring Figure 1 is correctly referenced.
>
> These corrections have been reflected in the updated PDF. We believe these changes significantly improve the readability and rigor of our paper.

---

> ### Author Response · Authors · 2025-11-27
> **Response Part 3**
>
> | Model            | Bo1   | Bo2   | Bo4   | Bo8   | Bo16  | Bo32  | Bo64  | Bo128 |
> | :--------------- | :---: | :---: | :---: | :---: | :---: | :---: | :---: | :---: |
> | **VRPRM-Qwen2.5** | **41.16** | **51.68** | **56.60** | **64.65** | **71.14** | **77.40** | **80.64** | **81.43** |
> | VRPRM-Qwen3-4B   | 41.16 | 49.89 | 55.48 | 63.31 | 70.25 | 75.17 | 78.08 | 79.42 |
> | MM-PRM           | 41.16 | 41.83 | 42.06 | 43.40 | 44.30 | 43.18 | 40.49 | 42.51 |
> | **Metric (K)**   | **1** | **2** | **4** | **8** | **16**| **32**| **64**| **128**|
> | Major@K          | 41.16 | 41.16 | 41.83 | 42.95 | 44.30 | 44.74 | 45.19 | 43.85 |
> | Pass@K           | 41.16 | 53.24 | 59.06 | 68.90 | 74.50 | 80.54 | 83.89 | 85.23 |
> Table 2: Best-of-N Results on Qwen2.5-VL-7B Policy, LogicVista
>
> | Model            | Bo1   | Bo2   | Bo4   | Bo8   | Bo16  | Bo32  | Bo64  | Bo128 |
> | :--------------- | :---: | :---: | :---: | :---: | :---: | :---: | :---: | :---: |
> | **VRPRM-Qwen2.5** | **37.44** | **42.01** | **48.10** | **52.79** | **54.82** | **57.36** | **60.15** | **62.06** |
> | VRPRM-Qwen3-4B   | 37.44 | 40.99 | 44.67 | 48.86 | 51.02 | 54.57 | 55.58 | 57.99 |
> | MM-PRM           | 37.44 | 39.09 | 40.74 | 42.51 | 42.13 | 42.77 | 40.99 | 40.86 |
> | **Metric (K)**   | **1** | **2** | **4** | **8** | **16**| **32**| **64**| **128**|
> | Major@K          | 37.44 | 37.94 | 38.07 | 40.23 | 41.37 | 42.13 | 42.01 | 42.26 |
> | Pass@K           | 37.44 | 44.42 | 52.54 | 57.23 | 62.31 | 65.86 | 67.64 | 69.29 |
> Table 3: Best-of-N Results on Qwen2.5-VL-7B Policy, MathVerse-VO
>
> | Model            | Bo1   | Bo2   | Bo4   | Bo8   | Bo16  | Bo32  | Bo64  | Bo128 |
> | :--------------- | :---: | :---: | :---: | :---: | :---: | :---: | :---: | :---: |
> | **VRPRM-Qwen2.5** | **64.00** | **67.70** | **70.40** | **72.20** | **74.90** | **76.80** | **79.30** | **81.50** |
> | VRPRM-Qwen3-4B   | 64.00 | 67.00 | 70.10 | 71.60 | 72.80 | 75.30 | 77.80 | 79.00 |
> | MM-PRM           | 64.00 | 63.80 | 62.90 | 62.70 | 63.10 | 63.50 | 64.30 | 64.20 |
> | **Metric (K)**   | **1** | **2** | **4** | **8** | **16**| **32**| **64**| **128**|
> | Major@K          | 64.00 | 64.00 | 63.60 | 63.70 | 64.20 | 64.70 | 64.90 | 64.80 |
> | Pass@K           | 64.00 | 69.40 | 73.40 | 76.30 | 78.00 | 80.80 | 83.00 | 84.20 |
> Table 4: Best-of-N Results on Qwen2.5-VL-7B Policy, MathVista
>
> | Model            | Bo1   | Bo2   | Bo4   | Bo8   | Bo16  | Bo32  | Bo64  | Bo128 |
> | :--------------- | :---: | :---: | :---: | :---: | :---: | :---: | :---: | :---: |
> | **VRPRM-Qwen2.5** | **24.90** | **30.56** | **37.66** | **44.18** | **50.30** | **55.72** | **60.76** | **64.31** |
> | VRPRM-Qwen3-4B   | 24.90 | 27.86 | 32.04 | 35.76 | 40.23 | 45.00 | 49.90 | 53.52 |
> | MM-PRM           | 24.90 | 25.86 | 26.41 | 27.07 | 27.01 | 26.97 | 27.20 | 25.89 |
> | **Metric (K)**   | **1** | **2** | **4** | **8** | **16**| **32**| **64**| **128**|
> | Major@K          | 24.90 | 24.97 | 25.99 | 27.07 | 27.60 | 27.50 | 27.73 | 27.50 |
> | Pass@K           | 24.90 | 32.43 | 40.26 | 47.63 | 54.47 | 59.80 | 64.70 | 69.08 |
> Table 5: Best-of-N Results on Qwen2.5-VL-7B Policy, MathVision

---

> > ### Author Response · Authors · 2025-12-03
> > **Response Part 4**
> >
> > | Model | BoN | LogicVista | MathVerse-VO | MathVista | MathVision |
> > | :--- | :--- | :--- | :--- | :--- | :--- |
> > | **Pass@K** | 1 | 36.38 | 22.80 | 64.50 | 17.00 |
> > | | 2 | 54.14 | 37.44 | 69.40 | 30.76 |
> > | | 4 | 72.26 | 48.98 | 77.50 | 43.98 |
> > | | 8 | 85.68 | 57.74 | 83.20 | 56.55 |
> > | | 16 | 92.62 | 65.48 | 87.90 | 68.75 |
> > | | 32 | 96.64 | 71.83 | 90.60 | 76.81 |
> > | | 64 | 98.21 | 76.14 | 92.90 | 82.34 |
> > | | 128 | 98.66 | 78.55 | 94.10 | 86.28 |
> > | **Major@K** | 1 | 36.38 | 22.80 | 64.50 | 17.00 |
> > | | 2 | 36.24 | 24.49 | 57.20 | 19.01 |
> > | | 4 | 41.61 | 27.16 | 62.10 | 20.92 |
> > | | 8 | 41.83 | 31.09 | 63.40 | 23.36 |
> > | | 16 | 43.18 | 32.61 | 65.10 | 25.92 |
> > | | 32 | 43.62 | 33.38 | 65.20 | 25.66 |
> > | | 64 | 42.95 | 33.88 | 65.00 | 26.38 |
> > | | 128 | 42.51 | 33.76 | 65.50 | 26.48 |
> > | **VisualPRM** | 1 | 36.38 | 22.80 | 64.50 | 17.00 |
> > | | 2 | 41.83 | 29.70 | 64.00 | 22.63 |
> > | | 4 | 40.49 | 31.85 | 67.30 | 24.18 |
> > | | 8 | 43.80 | 35.80 | 68.50 | 25.70 |
> > | | 16 | 42.50 | 36.40 | 69.90 | 27.30 |
> > | | 32 | 43.40 | 37.80 | 70.40 | 29.60 |
> > | | 64 | 45.40 | 38.20 | 69.60 | 30.60 |
> > | | 128 | 45.40 | 39.30 | 70.80 | 30.30 |
> > | **VRPRM w/o RL** | 1 | 36.38 | 22.80 | 64.50 | 17.00 |
> > | | 2 | 41.96 | 31.98 | 63.10 | 23.65 |
> > | | 4 | 52.01 | 37.44 | 67.70 | 28.42 |
> > | | 8 | 62.60 | 39.85 | 72.60 | 33.95 |
> > | | 16 | 64.06 | 43.53 | 74.20 | 37.11 |
> > | | 32 | 65.85 | 46.83 | 75.40 | 41.25 |
> > | | 64 | 70.54 | 49.75 | 75.30 | 45.26 |
> > | | 128 | 70.76 | 48.98 | 75.80 | 47.89 |
> > | **VRPRM** | 1 | 36.38 | 22.80 | 64.50 | 17.00 |
> > | | 2 | 47.32 | 35.15 | 66.60 | 28.09 |
> > | | 4 | 63.84 | 43.27 | 72.30 | 38.72 |
> > | | 8 | 79.46 | 51.52 | 79.10 | 51.44 |
> > | | 16 | 86.83 | 58.25 | 83.20 | 61.02 |
> > | | 32 | 91.52 | 63.32 | 86.60 | 69.57 |
> > | | 64 | 96.21 | 68.27 | 89.30 | 75.79 |
> > | | 128 | 96.54 | 69.54 | 90.60 | 80.13 |
> > | **VRPRM-MiMo** | 1 | 36.38 | 22.80 | 64.50 | 17.00 |
> > | | 2 | 49.44 | 32.49 | 67.70 | 27.34 |
> > | | 4 | 66.22 | 41.50 | 74.90 | 37.66 |
> > | | 8 | 77.63 | 50.38 | 81.60 | 49.77 |
> > | | 16 | 86.35 | 56.60 | 85.60 | 61.48 |
> > | | 32 | 90.83 | 63.07 | 88.10 | 71.09 |
> > | | 64 | 94.41 | 67.51 | 91.10 | 77.50 |
> > | | 128 | 94.63 | 71.07 | 92.60 | 82.47 |
> >
> > Table 6: Best-of-N Results on InternVL2.5-8B Policy

---

### Official Review · Reviewer_T1Q6 · 2025-10-31

**Soundness:** 1
**Presentation:** 2
**Contribution:** 3
**Rating:** 4
**Confidence:** 4

**Summary:**

The paper proposes VRPRM, a multimodal Process Reward Model trained with a two-stage scheme: (i) cold-start SFT on ~3.6K CoT-PRM data generated by Claude-3.7-Sonnet to teach formatting and step-wise judging, and (ii) RL scaling on 50K non-CoT PRM data. The PRM predicts a binary step label and uses the probability of token “1” at each step as the step reward (Eq. 9), then aggregates over steps for BoN selection. On VisualProcessBench (FEI/AEI) and multiple reasoning benchmarks, VRPRM reportedly outperforms VisualPRM and yields strong BoN gains with InternVL2.5 policies.

**Strengths:**

Clear, practical PRM pipeline: The two-stage training design (CoT-PRM SFT → RL on non-CoT) is well-motivated and carefully specified, with strict format/quality checks.

Strong empirical results on process evaluation: On VisualProcessBench, VRPRM-7B variants outperform prior work (incl. VisualPRM) on both FEI/AEI.

**Weaknesses:**

Cross-family generalization not fully established: Most experiments pair VRPRM with InternVL2.5 policies for test-time scaling, with no results on other families (e.g., Qwen-VL or GPT-class).

RL training dynamics are underreported: The paper does not show reward trajectories or response-length curves during RL, making it hard to diagnose the training process.

Higher inference overhead for CoT-PRM: Compared to PRMs that output a single {+/-} token, the CoT-PRM requires structured reasoning and stepwise judgments, which substantially increases per-candidate evaluation time.

**Questions:**

1. Please address the key weaknesses we raised.

2. Strong-policy stress test on HLE: In light of VRPRM’s strong performance with InternVL models, it would be very helpful to also report BoN results with a state-of-the-art non-InternVL policy (e.g., GPT-5-mini or a comparable model you can access) on Humanity’s Last Exam (HLE, with images), to see whether VRPRM could further advance SOTA.

3. Concrete example of VRPRM scoring:
Please include one full worked example (preferably from your test set) showing: (1) The policy’s generated candidate. The corresponding VRPRM's generated judgments. (2) For the selected candidate, the step list and the per-step probability of token “1” (your positive label).

---

> ### Author Response · Authors · 2025-11-27
> **Response Part 1**
>
> Dear Reviewer T1Q6:
>
> Thank you for your valuable comments. We appreciate the opportunity to clarify and address your concerns, which we believe will significantly enhance our paper. Below, we provide detailed responses to each point you raised:
> # W1:Response to Cross-family generalization not fully established
>
> We thank the reviewer for the insightful suggestion regarding cross-family generalization and stress testing on stronger policies. We agree that evaluating VRPRM beyond the InternVL family is crucial to demonstrating the transferability of our training paradigm.
>
> To address the concern that our performance might be specific to the InternVL series, we conducted extensive Best-of-N experiments using **Qwen2.5-VL-7B** as a new policy model. We evaluated VRPRM on four benchmarks, comparing it against the Reward Model Baseline (MM-PRM) and our two variants: VRPRM-Qwen2.5 and VRPRM-Qwen3 . We report `pass@k` (theoretical upper bound) and `major@k` (self-consistency baseline). The results are summarized in **Tables 1-4**.
>
> **Ovservation 1**: **Robust Generalization**. When applied to the Qwen2.5-VL-7B policy, our VRPRM models **consistently and significantly** outperform the MM-PRM baseline across all datasets. This confirms that the VRPRM training paradigm is **model-agnostic**; the process reward capability learned via our two-stage strategy effectively transfers from InternVL to the Qwen family without architecture-specific tuning.
>
> **Observation 2**: **Compatibility with Thinking Models**. The VRPRM trained on **Qwen3-4B-VL-Thinking** (a model recently published with intrinsic thinking capabilities) demonstrates comparable performance to our VRPRM-Qwen2.5. This indicates that our reward modeling approach is highly compatible with emerging thinking models, effectively guiding models with long reasoning chains to generate appropriate rewards.
>
> | Model            | Bo1   | Bo2   | Bo4   | Bo8   | Bo16  | Bo32  | Bo64  | Bo128 |
> | :--------------- | :---: | :---: | :---: | :---: | :---: | :---: | :---: | :---: |
> | **VRPRM-Qwen2.5** | **41.16** | **51.68** | **56.60** | **64.65** | **71.14** | **77.40** | **80.64** | **81.43** |
> | VRPRM-Qwen3-4B   | 41.16 | 49.89 | 55.48 | 63.31 | 70.25 | 75.17 | 78.08 | 79.42 |
> | MM-PRM           | 41.16 | 41.83 | 42.06 | 43.40 | 44.30 | 43.18 | 40.49 | 42.51 |
> | **Metric (K)**   | **1** | **2** | **4** | **8** | **16**| **32**| **64**| **128**|
> | Major@K          | 41.16 | 41.16 | 41.83 | 42.95 | 44.30 | 44.74 | 45.19 | 43.85 |
> | Pass@K           | 41.16 | 53.24 | 59.06 | 68.90 | 74.50 | 80.54 | 83.89 | 85.23 |
>
> Table 1: Best-of-N Results on Qwen2.5-VL-7B Policy, LogicVista
>
> | Model            | Bo1   | Bo2   | Bo4   | Bo8   | Bo16  | Bo32  | Bo64  | Bo128 |
> | :--------------- | :---: | :---: | :---: | :---: | :---: | :---: | :---: | :---: |
> | **VRPRM-Qwen2.5** | **37.44** | **42.01** | **48.10** | **52.79** | **54.82** | **57.36** | **60.15** | **62.06** |
> | VRPRM-Qwen3-4B   | 37.44 | 40.99 | 44.67 | 48.86 | 51.02 | 54.57 | 55.58 | 57.99 |
> | MM-PRM           | 37.44 | 39.09 | 40.74 | 42.51 | 42.13 | 42.77 | 40.99 | 40.86 |
> | **Metric (K)**   | **1** | **2** | **4** | **8** | **16**| **32**| **64**| **128**|
> | Major@K          | 37.44 | 37.94 | 38.07 | 40.23 | 41.37 | 42.13 | 42.01 | 42.26 |
> | Pass@K           | 37.44 | 44.42 | 52.54 | 57.23 | 62.31 | 65.86 | 67.64 | 69.29 |
>
> Table 2: Best-of-N Results on Qwen2.5-VL-7B Policy, MathVerse-VO
>
> | Model            | Bo1   | Bo2   | Bo4   | Bo8   | Bo16  | Bo32  | Bo64  | Bo128 |
> | :--------------- | :---: | :---: | :---: | :---: | :---: | :---: | :---: | :---: |
> | **VRPRM-Qwen2.5** | **64.00** | **67.70** | **70.40** | **72.20** | **74.90** | **76.80** | **79.30** | **81.50** |
> | VRPRM-Qwen3-4B   | 64.00 | 67.00 | 70.10 | 71.60 | 72.80 | 75.30 | 77.80 | 79.00 |
> | MM-PRM           | 64.00 | 63.80 | 62.90 | 62.70 | 63.10 | 63.50 | 64.30 | 64.20 |
> | **Metric (K)**   | **1** | **2** | **4** | **8** | **16**| **32**| **64**| **128**|
> | Major@K          | 64.00 | 64.00 | 63.60 | 63.70 | 64.20 | 64.70 | 64.90 | 64.80 |
> | Pass@K           | 64.00 | 69.40 | 73.40 | 76.30 | 78.00 | 80.80 | 83.00 | 84.20 |
>
> Table 3: Best-of-N Results on Qwen2.5-VL-7B Policy, MathVista
>
> | Model            | Bo1   | Bo2   | Bo4   | Bo8   | Bo16  | Bo32  | Bo64  | Bo128 |
> | :--------------- | :---: | :---: | :---: | :---: | :---: | :---: | :---: | :---: |
> | **VRPRM-Qwen2.5** | **24.90** | **30.56** | **37.66** | **44.18** | **50.30** | **55.72** | **60.76** | **64.31** |
> | VRPRM-Qwen3-4B   | 24.90 | 27.86 | 32.04 | 35.76 | 40.23 | 45.00 | 49.90 | 53.52 |
> | MM-PRM           | 24.90 | 25.86 | 26.41 | 27.07 | 27.01 | 26.97 | 27.20 | 25.89 |
> | **Metric (K)**   | **1** | **2** | **4** | **8** | **16**| **32**| **64**| **128**|
> | Major@K          | 24.90 | 24.97 | 25.99 | 27.07 | 27.60 | 27.50 | 27.73 | 27.50 |
> | Pass@K           | 24.90 | 32.43 | 40.26 | 47.63 | 54.47 | 59.80 | 64.70 | 69.08 |
>
> Table 4: Best-of-N Results on Qwen2.5-VL-7B Policy, MathVision

---

> ### Author Response · Authors · 2025-11-27
> **Response Part 2**
>
> # W2:Response to RL training dynamics are underreported
>
> We appreciate the reviewer's suggestion to incorporate training dynamics, as they are crucial for diagnosing the stability of the reinforcement learning process. In the revised manuscript (**Appendix**), we have included detailed training curves for VRPRM, specifically tracking `overall_reward`, `format_score`, `process_score`, and `response_length`.
>
> - **Reward Trajectories**: The curves exhibit a consistent upward trend followed by stabilization, indicating that the model effectively optimized the objective function. The optimization process remained stable without significant collapse or spikes, validating the effectiveness of the GRPO algorithm in our setting.
>
> - **Response Length**: The average length initially increased as the model learned to reason explicitly, then gradually stabilized. Collectively, these trajectories demonstrate that the training process of VRPRM is both healthy and stable.
>
> # W3:Response to Higher inference overhead for CoT-PRM
>
> We thank the reviewer for raising the concern regarding inference overhead. We acknowledge that generating structured reasoning steps inevitably increases latency compared to scalar-based reward models, and we conducted a quantitive analysis on the computation to further illustrate and justify the CoT.
>
> 1. **Quantitive Analysis:**
>
> We evaluated inference time and token consumption on the **VisualProcessBench(MMMU)** dataset using **NVIDIA H200 GPUs**. We compared VRPRM (Full CoT) against VRPRM (w/o CoT) and the VisualPRM baseline. The results are shown in **Table 5**. The inference time drops inevitably when the CoT module is disabled, confirming that the autoregressive CoT generation primarily contributes to the additional overhead.
>
> | Model           | Avg Total Tokens / Sample | Avg Output Tokens / Sample | AVG Input Tokens / Sample | AVG Time / (Sample*GPU) (s) | AVG Time / (Reward*GPU) (s)|
> |-----------------|---------------------------|----------------------------|---------------------------|-------------------------|-------------------------|
> | VRPRM-Qwen2.5   | 2305.82                   | 339.73                     | 1966.08                   | 16.94                   | 1.54                    |
> | - w/o CoT       | 1976.12                   | 10.03                      | 1966.08                   | 0.39                    | 0.04                    |
> | VisualPRM-8B    | 1344.88                   | 10.03                      | 1334.85                   | 0.07                    | 0.01                    |
>
> Table 5: Cost Analysis on VisualProcessBench(MMMU)
>
> 2. **Justification:**
>
> We argue that this computational cost is a necessary trade-off for the following reasons.
> - **Performance**. As shown in Table 6, disabling the CoT module for efficiency causes a noticible drop on the All Error Identification (AEI) and First Error Identification (FEI) score on VisualProcessBench. This implies that explicit reasoning is the core mechanism enabling the model's fine-grained error correction capability.
> - **Interpretability**. Unlike traditional models that only output a scalar score, VRPRM provides interpretable reasoning steps. This allows the model to explain why a step is incorrect, which is critical for verification tasks. We view the additional compute as the cost for acquiring this diagnostic capability.
>
> | Model         | AEI    | FEI    | AVG    |
> |---------------|--------|--------|--------|
> | VRPRM-Qwen2.5 | 63.16  | 52.06  | 57.61  |
> | - w/o CoT     | 52.66  | 46.44  | 49.55  |
> | VisualPRM-8B  | 59.01  | 30.71  | 44.86  |
>
> Table 6. AEI and FEI Results on VisualProcessBench(MMMU)

---

> ### Author Response · Authors · 2025-11-27
> **Response Part 3**
>
> # Q2:Response to Strong-policy stress test on HLE
>
> Following your suggestion, we successfully completed the stress test using **gpt-5-mini-2025-08-07** as the strong policy model. We evaluated VRPRM against the Self-Consistency baseline (`Major@K`) and the prior SOTA baseline (`VisualPRM-8B`). The results are presented below:
>
> | Model | Bo1 | Bo2 | Bo4 | Bo8 | Bo16 | Bo32 | Bo64 |
> |---|---|---|---|---|---|---|---|
> | Pass@K | 11.11 | 14.62 | 16.67 | 23.1 | 26.61 | 31.58 | 34.5 |
> | Major@K | 11.11 | 11.11 | 11.4 | **12.28** | **12.87** | 13.45 | 13.74 |
> | VisualPRM-8B | 11.11 | 10.53 | 10.23 | 10.53 | 10.82 | 10.82 | 10.82 |
> | VRPRM-Qwen2.5 | **11.11** | **11.4** | **11.99** | 11.11 | 11.4 | **13.45** | **14.04** |
>
> Table 7. Best-of-N Results on HLE (with images) (Policy: gpt-5-mini-2025-08-07)
>
> As shown in **Table 7**, the baseline `VisualPRM-8B` struggles significantly on this dataset, with performance stagnating or even dropping below the Bo1 baseline (10.82% vs 11.11%) as N increases. In contrast, `VRPRM` demonstrates a **positive scaling trend**, reaching **14.04%** at Bo64.
>
> Crucially, on extremely difficult tasks, reward models often struggle to outperform Self-Consistency (`Major@K`). However, at N=64, VRPRM (**14.04%**) successfully surpasses Major@K (**13.74%**). This confirms that our process reward signal provides **discriminative value beyond simple consensus**, effectively identifying correct answers even when they are rare "needles in a haystack."
>
> This experiment confirms that VRPRM's generalization capability extends to the most challenging multimodal reasoning scenarios currently available, outperforming both prior PRMs and voting baselines. We will include these results in the final version of the paper.
>
> # Q3:Response to Concrete example of VRPRM scoring
>
> We thank the reviewer for this suggestion. To demonstrate VRPRM’s fine-grained verification capability, we provide a case study from the **LogicVista** test set (Policy: **Qwen2.5-VL-7B**, Judge: **VRPRM-Qwen2.5**).
>
> **Task**: Identify which set a figure belongs to based on petal colors.
>
> **Ground Truth**: Set B (Visual feature: "White outside / Black inside").
>
> We compare VRPRM's evaluation of two candidate responses:
>
> 1. **Candidate A (Correct & Selected)**
>
> **Policy Step**: "...petals are white on the outside and black on the inside... matches Set B."
>
> **VRPRM CoT**: "This step accurately describes the characteristics... This is factually correct based on the image."
>
> **VRPRM Scores**: [1.0, 1.0, 1.0, 1.0, 1.0]
>
> **Outcome: Preferred (Avg Score = 1.0)**
>
> 2. **Candidate B (Hallucination & Rejected)**
>
> **Policy Step 2 (The Error)**: "...petals are **black on the outside and white on the inside**, which matches Set A." (Visual Hallucination)
>
> **VRPRM CoT (The Detection)**: "...This step correctly identifies that the figure... matches Set A. **However, this is incorrect because the figure actually matches Set B's description. \boxed{0}**"
>
> **VRPRM Scores**: [1.0, 1.0, 5.22e-05, 1.81e-07, 3.29e-06]
>
> **Outcome: Rejected (Avg Score = 0.4)**
>
> **Conclusion**: This example demonstrates that VRPRM does not act as a black-box scorer. By utilizing **Visual CoT**, it successfully grounds the text in the image, pinpointing exactly where the policy hallucinates (Step 2) and penalizing the score immediately. We will include the full transcript of this case in the **Appendix**.

---

### Official Review · Reviewer_bYma · 2025-11-03

**Soundness:** 3
**Presentation:** 3
**Contribution:** 2
**Rating:** 6
**Confidence:** 3

**Summary:**

This paper introduces VRPRM (Visual Reasoning Process Reward Model), a new Process Reward Model (PRM) designed to perform fine-grained, step-by-step evaluation of long-term multimodal reasoning processes. The work addresses the limitation of existing PRMs, which often lack the capability for deep, Chain-of-Thought (CoT) reasoning, especially in multimodal settings. VRPRM proposes an efficient two-stage training strategy: 1) initial Supervised Fine-Tuning (SFT) on a small amount of CoT-PRM data (3.6K samples) to establish reasoning foundations, followed by 2) Reinforcement Learning (RL) fine-tuning (GRPO) on a larger set of non-CoT PRM data (50K samples) to enhance robustness and scalability. The model demonstrates strong performance, with results indicating it can significantly surpass non-thinking PRMs trained on much larger datasets, achieving a high relative performance improvement over the base model in the BoN (Best-of-N) experiment.

**Strengths:**

1. **Effective Multi-stage Training for Data Efficiency:** The core methodology, combining a small, high-quality SFT CoT dataset with a larger, less costly RL dataset, is highly practical. The results strongly suggest that the initial SFT phase effectively primes the model for complex reasoning, allowing the subsequent RL stage to generalize this capability robustly and efficiently, overcoming the high annotation cost typically associated with CoT-PRM data.

2. **Demonstrated Performance Gains:** The model achieves impressive empirical results, notably achieving a relative performance improvement of up to 118% over the base model in the BoN experiment and surpassing non-thinking PRMs trained on 400K data points using only a fraction of that data. This validates the effectiveness of the process reward modeling approach guided by visual reasoning.

3. **Relevant Application of Existing Techniques:** The paper successfully adapts established and robust techniques—specifically SFT and GRPO (General Reinforcement Policy Optimization)—to the multimodal PRM domain, providing a clean, reproducible framework for advancing step-wise reward modeling in VLMs. The design of the PRM response format to capture step-by-step judgments is clear and effective.

**Weaknesses:**

1. **Increased Computational Overhead:** The explicit nature of the Process Reward Model, which generates a step-by-step judgment (a reasoning trace of its own) for the solver's output, inevitably introduces significant computational overhead compared to non-reasoning, end-to-end reward models (which only generate a single score). The paper does not provide a quantitative analysis of this overhead (e.g., token generation time or total inference latency) relative to simpler PRM or reward model baselines.

2. **Limited Technical Novelty in Methodology:** The core technical components of the method—Supervised Fine-Tuning (SFT) followed by GRPO (a common RL fine-tuning variant)—are standard practices in LLM post-training. The paper's novelty lies primarily in its successful application and empirical demonstration on Multimodal PRMs with a specific, designed output format (process judgment blocks), rather than in the development of a fundamentally PRM-specific new training algorithm.

3. **Unverified SFT Data Quality:** The CoT-PRM SFT dataset, which is critical for establishing the model's fundamental reasoning capability, is generated by a powerful proprietary model (Claude) and only subjected to a format check. There is a significant concern that the step-wise correctness or fine-grained judgment labels assigned by Claude may contain subtle errors or biases that are not caught by a simple format validation. This introduces uncertainty regarding the true quality and correctness of the most expensive and foundational data used in the training process.

**Questions:**

1. Verification of SFT Step-Labels: Given the reliance on Claude for generating the CoT-PRM SFT data, what verification process, beyond a simple format check, was used to ensure the correctness and accuracy of the step-wise judgment labels themselves?

2. Quantitative Overhead Analysis: Given that the VRPRM generates a detailed step-wise reasoning trace, could the authors provide a quantitative analysis of the computational overhead?

3. Reward Model  Generalization: How robust is the final model's step-wise judgment when evaluating reasoning traces for unseen or out-of-distribution multimodal tasks?

---

> ### Author Response · Authors · 2025-11-27
> **Response Part 1**
>
> Dear Reviewer bYma:
> We sincerely thank you for recognizing our work as comprehensive and identifying an important research direction. We value your constructive comments and suggestions. Below, we provide detailed responses to clarify the points raised and address your concerns.
> # W1&Q2: Response to Quantitative Overhead Analysis
> 1. **Quantitative Analysis**
>
> We thank the reviewer for the constructive feedback regarding computational overhead. We acknowledge that introducing the Chain-of-Thought (CoT) mechanism increases inference latency compared to non-reasoning reward models. To address this, we conducted a quantitative analysis on the **VisualProcessBench (MMMU)** dataset. We recorded the average token consumption and inference time per sample for VRPRM-Qwen2.5 (in both CoT and non-CoT modes) and the baseline VisualPRM-8B. All computations were performed on NVIDIA H200 GPUs. The results are summarized in **Table 1**:
>
> | Model           | Avg Total Tokens / Sample | Avg Output Tokens / Sample | AVG Input Tokens / Sample | AVG Time / (Sample*GPU) (s) | AVG Time / (Reward*GPU) (s)|
> |-----------------|---------------------------|----------------------------|---------------------------|-------------------------|-------------------------|
> | VRPRM-Qwen2.5   | 2305.82                   | 339.73                     | 1966.08                   | 16.94                   | 1.54                    |
> | - w/o CoT       | 1976.12                   | **10.03**                      | 1966.08                   | 0.39                    | 0.04                    |
> | VisualPRM-8B    | **1344.88**                   | **10.03**                      | **1334.85**                   | **0.07**                    | **0.01**                    |
>
> Table 1: Cost Analysis on VisualProcessBench(MMMU)
>
> **Observation**: The inference time for VRPRM drops significantly from 16.94s to 0.39s when the CoT module is disabled, bringing it closer to the baseline's magnitude (0.07s). This confirms that the primary computational burden stems from the autoregressive generation of the reasoning trace.
>
> 2. **Justification of Necessity**
>
> We argue that this increased test-time computation is a necessary investment for two specific reasons.
> - **Performance Gain.** The explicit reasoning process is the core mechanism enabling fine-grained error correction. We evaluated the fine-grained First Error Identification and All Error Identification results on **VisualProcessBench(MMMU)**. As shown in **Table 2**, removing the CoT component leads to a substantial degradation in performance.
> - **Interpretability.** Unlike traditional reward models that only output scalar scores, VRPRM provides transparent reasoning processes. We regard the computational overhead as the cost of acquiring this white-box diagnostic capability, transforming VRPRM from a simple scorer into a trustworthy verifier.
>
> | Model         | AEI    | FEI    | AVG    |
> |---------------|--------|--------|--------|
> | VRPRM-Qwen2.5 | **63.16**  | **52.06**  | **57.61**  |
> | - w/o CoT     | 52.66  | 46.44  | 49.55  |
> | - w/o RL      | 55.94  | 47.57  | 51.76  |
> | VisualPRM-8B  | 59.01  | 30.71  | 44.86  |
>
> Table 2. AEI and FEI Results on VisualProcessBench(MMMU)
>
> # W2: Response to Limited Technical Novelty in Methodology
>
> We understand the reviewer's concern regarding the technical novelty of the SFT + GRPO training components. However, we respectfully submit that our innovation lies in the **strategic design of the training paradigm and data utilization**, rather than merely applying existing algorithms.
> - We are the first to demonstrate within the multimodal PRM domain that explicit reasoning (Chain-of-Thought) capabilities can be reinforced during the RL phase using only standard step-level labels (non-CoT data).
> - VRPRM achieves SOTA performance using less than 1/8 of the training data compared to baselines. This proves that our approach is technically superior to the large-scale SFT methods relied upon in prior work.
> - VRPRM is the first multimodal PRM to integrate CoT reasoning with RL optimization. Previous visual PRMs (e.g., VisualPRM, Qwen-Math-PRM) are limited to scoring without explaining their judgments, leading to poor generalization on complex reasoning tasks. While GenPRM possesses reasoning capabilities, it lacks RL reinforcement, resulting in high data costs and limited reasoning potential.

---

> ### Author Response · Authors · 2025-11-27
> **Response Part 2**
>
> # W3&Q1: Response to Unverified SFT Data Quality
>
> We appreciate the reviewer's detailed examination of the quality of our synthetic SFT data. However, we would like to clarify a critical misunderstanding regarding our data filtering pipeline. The reviewer stated that the data was "only checked for format." This is not the case.
> As described in the second paragraph of **Section 3.2.1** ("Step 2, we then guide the model to perform a fine-grained analysis of each solution step and annotate the correctness of each step in a unified format, in the form of $\backslash$boxed\{1\} (correct) or $\backslash$boxed\{0\} (incorrect). Step 3, the model must also return the intermediate results of the evaluation process in a standardized JSON format, such as \{``Score'':[$r_1,\dots,r_n$]\};") and illustrated in the "Quality Inspection" module of **Figure 3**, our filtering process is significantly more rigorous:
> 1. The source dataset we utilized, VisualPRM400K, already contains correct ground-truth process labels. During data synthesis, we instruct Claude to generate both the reasoning process and the correctness judgment for each step. We then **strictly discard** any response where Claude’s predicted labels do not match the original reference labels.
> 2. We utilize Claude-3.7-Sonnet, widely recognized as one of the most powerful reasoning models available, known for its ability to generate high-quality Chain-of-Thought (CoT) explanations. By combining this powerful generator with the strict consistency filtering mentioned above, we effectively minimize the risk of hallucinations or erroneous reasoning logic.
> 3. Even assuming a small amount of noise remains in the synthetic data, our two-stage training strategy is designed to mitigate this. The SFT stage serves merely as a "cold start" to initialize the format and basic reasoning style. The subsequent Reinforcement Learning (RL) stage utilizes process rewards to further refine the model. This is supported by the experimental results in **Table 2**: our model significantly outperforms baselines, which would not be possible if the SFT data contained fundamental flaws or biases preventing effective generalization to the test set.
>
> # Q3: Response to Reward Model Generalization
> We appreciate the reviewer’s critical question regarding generalization capabilities. We contend that VRPRM demonstrates exceptional robustness and cross-task transferability, significantly outperforming baselines across diverse and out-of-distribution (OOD) multimodal tasks. We support this argument with the following key evidence from our experiments:
> 1. Our evaluation is not confined to in-distribution data. We tested VRPRM on five distinct benchmarks (MathVista, MathVision, MathVerse, WeMath, and LogicVista), which vary significantly in terms of difficulty, visual complexity, and domain coverage. As demonstrated in **Table 3**, VRPRM achieved consistent improvements across all benchmarks compared to the VisualPRM baseline. Notably, the LogicVista dataset—which focuses on logical reasoning—presents a significant distributional shift from the standard mathematics and science problems prevalent in our training set. On this task, VRPRM achieved a remarkable relative improvement of 43.36% (improving from 39.64 to 83.00 with InternVL2.5-26B). This indicates that VRPRM has acquired generalized reasoning and verification capabilities that extend beyond the specific mathematical and physical domains encountered during training.
> 2. We attribute VRPRM’s success on unseen tasks to our core methodology: explicit visual reasoning. Traditional visual PRMs operate as black-box scorers, often overfitting to the specific content distributions of the training set. If a model cannot inherently understand or solve an out-of-distribution task, it struggles to score it accurately, thereby hindering generalization. In contrast, VRPRM adopts a "Think-then-Judge" mechanism. This ensures that evaluation is grounded in a genuine understanding of the task rather than simple pattern matching, significantly enhancing the model's robustness and generalization capabilities on previously unseen benchmarks.
>
> | Model           | MathVista | MathVision | MathVerse-VO | WeMath | LogicVista | Overall |
> |-----------------|-----------|------------|--------------|--------|------------|---------|
> | InternVL2.5-26B | 68.20     | 23.40      | 24.00        | 30.90  | 39.64      | 37.23   |
> | +VisualPRM      | 73.10     | 29.60      | 39.10        | 40.80  | 51.00      | 46.72   |
> | **+VRPRM**          | **81.20**     | **55.79**      | **53.55**        | **40.14**  | **83.00**      | **62.74**   |
>
> Table 3. The Results of Bo8 on five multimodal reasoning benchmarks.

---

### Author Response · Authors · 2025-11-27
**General Response: Thanks, Contributions, Common Concerns Across Multiple Reviewers**

We sincerely thank the reviewers for their constructive feedback and for recognizing VRPRM’s **data-efficient two-stage training paradigm** [bYma, jxEj], **strong empirical performance** (up to 118% relative improvement) [bYma, T1Q6, SHdC], and **practical scalability** [SHdC]. We are encouraged by the consensus that our framework provides a clean, reproducible path for advancing step-wise reward modeling.
Below, we address the three common concerns raised across the reviews.
# Common Concerns Across Multiple Reviewers
We thank the reviewers for their constructive feedback. Below, we address common concerns in a unified response, followed by detailed point-by-point replies.
## Methodological Novelty & Positioning [bYma, SHdC, jxEj]
VRPRM is the first multimodal CoT-PRM trained through reinforcement learning. It surpasses existing state-of-the-art models with extremely low annotation costs through a data-efficient two-stage training strategy and achieves a relative performance improvement of up to 118% as a powerful test-time scaling method.
- **First CoT-RL Integration for MLLMs**: VRPRM is the first multimodal CoT-PRM trained through reinforcement learning. By utilizing just 3.6K CoT-PRM samples for a 'cold start', the model (`- w/o RL` in **Table 1**) achieves 62.11 and the base model is 45.36 (`Qwen2.5-VL-7B` in **Table 1**). In the RL stage, we leverage non-CoT PRM data to implicitly reinforce reasoning, further boosting performance from 62.11 to 66.00 (`VRPRM-Qwen-7B` in **Table 1**). This confirms that RL can enhance process supervision without relying on expensive explicit CoT annotations.
- **Solving the Multimodal Data Bottleneck**: Bridging text methods to MLLMs is limited by the high cost of visual reasoning data. As shown in **Table 1**, VRPRM breaks this bottleneck: using 1/8th of the training data (53.6K vs. 400K) compared to the `VisualPRM` baseline, `VRPRM-Qwen-7B` significantly surpasses it (AEI 66.00 vs. 61.03). This confirms that scaling with high-quality 'Cold Start' data (3.6K) and 'RL' data (50K) is a superior solution to massive SFT.

| Model Name | # Samples | MMMU | MathVision | MathVerse-VO | DynaMath | WeMath | AEI Avg. |
| :--- | :---: | :---: | :---: | :---: | :---: | :---: | :---: |
| Qwen2.5-VL-7B | unk | 46.88 | 39.54 | 42.75 | 52.89 | 44.76 | 45.36 |
| VisualPRM-8B | 400K | 59.01 | 62.91 | 60.93 | 62.08 | 60.22 | 61.03 |
| VRPRM-Qwen-7B | 53.6K | **63.16** | **67.34** | **63.80** | **67.95** | **67.76** | **66.00** |
| - w/o CoT | 53.6K | 52.66 | 51.95 | 54.72 | 53.06 | 55.90 | 53.66 |
| - w/o RL | 3.6K | 55.94 | 61.82 | 62.43 | 63.08 | 67.30 | 62.11 |
| - w/o RL & w/o CoT | 3.6K | 50.69 | 54.57 | 51.70 | 55.26 | 48.79 | 52.20 |

Table 1: VisualProcessBench results reported with AEI.

## Robustness & Generalization [bYma, T1Q6, SHdC]
We conducted additional experiments to demonstrate that VRPRM is robust, model-agnostic, and applicable to broad domains:
- **OOD Tasks**: As shown in **Table 2**, on LogicVista, a benchmark focused on pure logical reasoning (distinct from our math-heavy training data), `VRPRM` achieves a massive +43.36 (39.64 -> 83.00) point improvement over the `InternVL2.5-26B` baseline. This significantly outperforms the `VisualPRM` baseline (51.00), proving that our model transfers effectively to unseen reasoning tasks.
- **Cross-Model Generalization**: To verify that our method is not architecture-specific, we tested Qwen2.5-VL-7B as a policy model (**Table 3**). `VRPRM-Qwen2.5` boosts the Overall score to 59.52, consistently outperforming both the base policy (41.88) and the `MM-PRM` baseline (43.92). Furthermore, `VRPRM-Qwen3` (trained on Qwen3-4B-VL-Thinking) also shows strong performance (54.88), confirming compatibility with diverse architectures.
- **Broad Scope**: As shown in **Table 1**, On MMMU, which covers diverse disciplines including Art, Medicine, and Humanities, `VRPRM-Qwen-7B` achieves an AEI of 63.16. This surpasses the `VisualPRM-8B` baseline (59.01), confirming that our reasoning capabilities extend well beyond pure mathematics.

| Model | MathVista | MathVision | MathVerse-VO | LogicVista | Overall |
|---|---|---|---|---|---|
| InternVL2.5-26B | 68.20  | 23.40  | 24.00  | 39.64  | 37.23  |
| VisualPRM | 73.10  | 29.60  | 39.10  | 51.00  | 46.72  |
| VRPRM w/o RL | 77.40  | 37.99  | 44.29  | 68.90  | 55.47  |
| VRPRM | **81.20**  | **55.79**  | **53.55**  | **83.00**  | **62.74** |

Table 2: Bo8 results on multimodal reasoning benchmarks, InternVL2.5-26B as policy model

| Model | MathVista | MathVision | MathVerse-VO | LogicVista | Overall |
|---|---|---|---|---|---|
| Qwen2.5-VL-7B | 64.00  | 24.90  | 37.44  | 41.16  | 41.88  |
| MM-PRM | 62.70  | 27.07  | 42.51  | 43.40  | 43.92  |
| VRPRM-Qwen3 | 71.60  | 35.76  | 48.86  | 63.31  | 54.88  |
| VRPRM-Qwen2.5 | **72.20**  | **44.18**  | **52.79**  | **68.90**  | **59.52** |

Table 3: Bo8 Results on multimodal reasoning benchmarks, Qwen2.5-VL-7B as policy model

---

> ### Author Response · Authors · 2025-12-03
> **General Response Part 2**
>
> ## Computational Overhead & Efficiency Analysis [bYma, T1Q6]
> We acknowledge the inference cost of CoT generation but justify it as a necessary trade-off for fine-grained correction and interpretability. We have conducted a detailed quantitative analysis and we clarify the trade-off behind this design choice from two perspectives: performance gain and interpretability.
> 1. **Quantification of Overhead** To make a quantitive analysis of the overhead, we benchmarked the inference latency and tokens of reward models on the VisualProcessBench(MMMU) dataset. Results in **Table 4** confirm that the primary computational cost stems directly from generating detailed reasoning traces.
>
> | Model           | Avg Total Tokens | Avg Output Tokens | AVG Input Tokens | AVG Time |
> |-----------------|------------------|-------------------|------------------|----------|
> | VRPRM-Qwen2.5   | 2305.82          | 339.73            | 1966.08          | 16.94    |
> | - w/o CoT       | 1976.12          | 10.03             | 1966.08          | 0.39     |
> | VisualPRM-8B    | 1344.88          | 10.03             | 1334.85          | 0.07     |
>
> Table 4: Cost Analysis on VisualProcessBench(MMMU)
>
> | Model         | AEI    | FEI    | AVG    |
> |---------------|--------|--------|--------|
> | VisualPRM-8B  | 59.01  | 30.71  | 44.86  |
> | VRPRM-Qwen2.5 | **63.16**  | **52.06**  | **57.61**  |
> | - w/o CoT     | 52.66  | 46.44  | 49.55  |
>
> Table 5. Performance Impact of CoT (VisualProcessBench(MMMU))
>
> 2. **Justification of the Trade-off** We argue that this increased "test-time compute" is expected and necessary for three critical reasons:
> - **Necessary for Fine-grained Error Correction**: Ablation studies (**Table 5**) reveal that removing CoT (w/o CoT) causes significant drops in AEI and FEI. This implies that explicit reasoning is the core mechanism enabling the model's fine-grained error correction capability.
> - **From Black-box to White-box (Interpretability)**: Unlike traditional reward models that only output scalar scores, VRPRM provides transparent reasoning processes. We view the computational overhead as the cost of acquiring this white-box diagnostic capability, transforming VRPRM from a simple scorer into a trustworthy verifier.
>
> - **Alignment with Frontier Industry Practices**: Using reasoning models for evaluation is emerging as a standard for scaling supervision. For instance, the recent release of **Grok 4.1** [3] explicitly states: "In order to optimize these non-verifiable reward signals, we developed new methods that let us use frontier agentic reasoning models as reward models to autonomously evaluate and iterate on responses at scale." VRPRM aligns with this paradigm, demonstrating that allocating compute to the reward model's reasoning process is essential for achieving SOTA performance in complex reasoning tasks.
>
> [1] VisualPRM: An Effective Process Reward Model for Multimodal Reasoning
>
> [2] The Lessons of Developing Process Reward Models in Mathematical Reasoning
>
> [3] xAI. (2025). Grok 4.1. https://x.ai/news/grok-4-1

---

### Meta-Review · Area_Chair_82ij · 2026-01-07

**Summary:**

This work proposes VRPRM, a multimodal Process Reward Model trained with a two-stage scheme. The work first uses SFT on a small, structured CoT-PRM dataset to seed reasoning, then applies RL on larger non-CoT PRM data to scale. The core technical components of the method seem to be standard practices in LLM post-training. The techniques are well-explored in the text-only domain - neither CoT-PRM nor RL is novel. The paper also omits discussion of some relevant works.

**Reviewer Concerns:**

The core technical components of the method seem to be standard practices in LLM post-training. The techniques are well-explored in the text-only domain - neither CoT-PRM nor RL is novel. The paper also omits discussion of some relevant works.  The concerns about novelties are not well addressed.

**Reviewer Scores:**

reviewers are likely to keep the rating.

---

### Decision · Program_Chairs · 2026-01-26

Reject